# Training an Ising machine with equilibrium propagation

Jérémie Laydevant [1] ✉, Danijela Marković [1] & Julie Grollier [1] ✉

Ising machines, which are hardware implementations of the Ising model of coupled spins, have been influential in the development of unsupervised learning algorithms at the origins of Artificial Intelligence (AI). However, their application to AI has been limited due to the complexities in matching supervised training methods with Ising machine physics, even though these methods are essential for achieving high accuracy. In this study, we demonstrate an efficient approach to train Ising machines in a supervised way through the Equilibrium Propagation algorithm, achieving comparable results to software-based implementations. We employ the quantum annealing procedure of the D-Wave Ising machine to train a fully-connected neural network on the MNIST dataset. Furthermore, we demonstrate that the machine's connectivity supports convolution operations, enabling the training of a compact convolutional network with minimal spins per neuron. Our findings establish Ising machines as a promising trainable hardware platform for AI, with the potential to enhance machine learning applications.

Investigating physical systems that can execute cognitive tasks based on their dynamic behaviors or statistical features has long been a topic of interest in physics, motivated by the quest to unravel the brain's learning capabilities[1]. The Ising system[2] of coupled spins, described by the Ising energy function:

$$E_{Ising} = \sum_{i>j} J_{ij}\sigma_i\sigma_j + \sum_i h_i\sigma_i, \tag{1}$$

has played a significant role in these developments[3–6]. We use here an Ising energy that differs from the standard one by a minus sign, in order to match the D-Wave formulation[7], and to be consistent with all the equations and learning rules in the rest of the paper. It can indeed be likened to a neural network, where the state of a spin $\sigma_i$ (up or down) corresponds to the activity of a binary neuron, the value of the coupling between spins $J_{ij}$ corresponds to the strength of the synaptic connection between the neurons they emulate, and the bias fields $h_i$ applied to individual spins correspond to the biases of the artificial neurons. The majority of learning demonstrations on Ising machines[8–14] have focused on implementing Boltzmann machines methods[15,16]. This algorithm capitalizes on the properties of physical systems, characterized by an energy function, to evolve towards an equilibrium state governed by Boltzmann statistics. However, Boltzmann machines are generative models that do not directly optimize a cost function related to a classification error. Furthermore, the parameters are typically evolved with approximations of the gradient, as the exact value is complicated and lengthy to compute. Boltzmann machines therefore underperform in difficult classification tasks when compared to standard supervised learning algorithms like backpropagation[17].

The recent boom in AI, driven by the advent of these highly effective supervised algorithms, has led to the development of various new hardware platforms for AI applications. These platforms aim to address the growing power consumption and computational demands associated with both training and inference phases in AI systems. Emerging AI hardware solutions exploit local memory and leverage the unique physical phenomena exhibited by novel components[1,18,19]. However, these new platforms face compatibility challenges with the most efficient supervised training methods, which rely on the minimization of a global cost function, such as error backpropagation. This is due to the intrinsically non-local nature of these methods and the fact that the calculation of associated gradients is based on mathematical procedures that do not correspond to the physics of the emerging devices used. As a result, a significant research effort is

[1]Laboratoire Albert Fert, CNRS, Thales, Université Paris-Saclay, 91767 Palaiseau, France. ✉e-mail: jeremie.laydevant@gmail.com; julie.grollier@cnrs-thales.fr

currently underway to develop algorithms capable of training these novel, AI-dedicated hardware platforms efficiently[20-27].

Our work belongs to the growing field of Physical neural networks[26], where the goal is to develop physical systems based on unconventional nanodevices that solve AI tasks through the natural laws of physics[1,28-30]. We aim to show that a physical system of coupled spins[7,30-41] can learn to perform supervised AI tasks through an algorithm that harnesses its intrinsic ability to minimize an energy, arising from the natural laws that govern our physical world.

Introduced in 2017, Equilibrium Propagation (EP)[42] has garnered significant attention for its ability to train in a supervised way convergent recurrent energy-based models. Unlike traditional methods that compute gradients of the objective function using Backpropagation Through Time (BPTT), EP employs a local learning rule that not only approximates BPTT-derived gradients[43] but also overcomes the limitations of conventional training in physical systems[22,23,44,45]. The algorithm requires that the physical system evolves toward a stable equilibrium state—that does not need to be the ground state[42]—through the minimization of an energy function such as:

$$E_{EP} = \frac{1}{2}\sum_i s_i^2 - \sum_{i>j} W_{ij}\rho(s_i)\rho(s_j) - \sum_i b_i\rho(s_i), \qquad (2)$$

where $s_i$, $s_j$ are the real and continuous states of the neurons, $W_{ij}$ are the symmetric synaptic weights connecting neurons $i$ and $j$, $b_i$ are individual biases applied to the neurons, and $\rho$ is a non-linear activation function such as *tanh*. The first term in Eq. (2) is a damping term that allows the system to reach a stable equilibrium state. In Equilibrium Propagation, the inference is performed by conditioning the steady state of the system with input values (free phase), while learning is achieved by dynamically perturbing the outputs to align them with the desired values (nudge phase) and thus minimize the objective loss function $\mathcal{L}$. The parameter changes required for learning are derived from local measurements of the equilibrium states, as opposed to a complex non-local analytic mathematical procedure like backpropagation. The corresponding learning rule for synaptic weights writes:

$$-\frac{\partial \mathcal{L}}{\partial W_{ij}} = \Delta W_{ij} \propto \frac{1}{\beta}\left[(\rho(s_i)\rho(s_j))^{*,nudge} - (\rho(s_i)\rho(s_j))^{*free}\right], \qquad (3)$$

with a term $\beta$ that characterizes the strength of the nudging force. It favors the equilibrium state obtained after the nudge phase, whose outputs are closer to the target value, and destabilizes the one obtained after the free phase. This is accomplished by decreasing the energy of the nudge state and increasing the energy of the free state. Numerical simulations of Equilibrium Propagation have demonstrated state-of-the-art performance on benchmark tasks such as MNIST[42,43], CIFAR-10[46], and Image-net-32[47], using both fully-connected networks and convolutional architectures.

Equilibrium Propagation is, therefore, an excellent candidate for training physical systems described by an energy function[44,45]. Ising Machines[48] are analog[32,37,39,49] or digital hardware[34,41] systems that are particularly fitted for this purpose, as they are designed to find the ground state of the Ising spin model. Moreover, they offer thousands of spins, and their reconfigurable coupling parameters facilitate training. However, their applications are mostly limited to solving combinatorial problems with fixed parameters[50]. Training Ising machines using Equilibrium Propagation would broaden their scope to supervised classification tasks and leverage their adjustable parameters. Nonetheless, three fundamental differences exist between the Ising model (Eq. (1)) and the EP model (Eq. (2)), which present challenges for training.

First, the Ising energy function (Eq. (1)) lacks the damping term present in the Equilibrium Propagation (EP) model (Eq. (2)), which allows the latter to reach steady state equilibrium intrinsically. Ising machines can approach the ground state using various annealing methods[7,39,51] or minimum gain principle[52], but destabilizing it for the nudge phase in EP remains challenging. Developing methods to gently manipulate the equilibrium state is necessary.

Second, the difference between Ising spins' two-state nature and EP's continuous-state neurons poses a challenge. Approaches must be devised to create smooth modifications of the spin system by the outputs, emulating the gradual changes in a neural network's learning process. This would bridge the gap between the two-state Ising machine and continuous-state EP neurons, enabling efficient training.

Third, the implementation of EP on Ising machines necessitates navigating the balance between connectivity and parallelism. In contrast to biological neural networks, which are highly interconnected systems where neurons evolve simultaneously, Ising machines typically fall into two distinct categories. The first category offers full connectivity but operates with sequential dynamics, leveraging measurement-feedback mechanisms for simulating the spin dynamics[32,52]. Conversely, the second category showcases fully parallel dynamics but is limited by its sparser physical connections[53]. While the former is preferred for combinatorial optimization in current applications, the natural parallel dynamics toward an equilibrium state in the latter is especially fitting for Equilibrium Propagation. Strategies then need to be established to adapt the network architecture to the Ising hardware's connectivity.

In this study, we report a critical advancement towards utilizing Ising machines for machine learning applications. Employing the commercial D-Wave Ising machine[7], composed of thousands of two-state components, and the Equilibrium Propagation algorithm[42], we successfully recognize handwritten digits from the MNIST/100 database[54], achieving recognition rates comparable to a fully connected network trained using software simulations on standard digital hardware.

We train the Ising machine with Equilibrium Propagation, taking advantage of its capacity to reach the ground state of its energy function through annealing during the free phase and reverse-annealing during the nudge phase. While the coupling between the spins of the D-Wave machine has a high precision (5–6 bits) approaching the full precision synapses of the original Equilibrium Propagation model[23,46], the spins of the machine correspond to neurons with binary activations. In order to train this binary system, we adapt procedures developed for training binary neural networks[55,56] and increase the number of outputs.

Finally, we demonstrate that the connectivity between near-neighbor spins on the DW-2000 chip, featuring the Chimera architecture[7], is inherently compatible with convolutional operations. We successfully train a compact convolutional network entirely on the chip, achieving recognition rates on par with software performance.

Our work belongs to the growing field of Physical neural networks[26], where the goal is to develop physical systems based on unconventional nanodevices that solve AI tasks through the natural laws of physics[1,28-30]. Our results show that a physical system of coupled spins can learn to perform supervised AI tasks through an algorithm that harnesses its intrinsic ability to minimize an energy, arising from the natural laws that govern our physical world.

Our results indicate that Ising machines hold significant potential as machine learning hardware, with their physics allowing for inference, error backpropagation, and gradient computation. Furthermore, our findings highlight the promise of physics-based learning algorithms, such as Equilibrium Propagation, in training fully connected and convolutional networks on emerging hardware.

## Results
### Training an Ising machine with EP through annealing
In this study, we have chosen the commercial Ising machine D-Wave as the demonstration platform for our algorithm, owing to its distinct

advantages compared to other publicly available IMs. Specifically, D-Wave offers a large number of spins, ranging from 2000-5000 depending on the version (2000Q or Advantage 4–5), high-precision coupling parameters (4–6 bits), and the ability to control these parameters online through a Python interface. This interface is fully compatible with the code developed for the training algorithm, which is essential, as the parameters need iterative adjustments during the training process.

We illustrate the implemented training procedure in Fig. 1a and provide a comprehensive description of how we execute the free phase, nudge phase, and learning rule in the subsequent sections. For both the free and nudge phases, we introduce the input data to the spin system through bias fields. In the following sections, we elaborate on how the bias fields are set according to the input data and the architecture implemented on the chip.

In each phase, the input bias fields are maintained constant, allowing the system to stabilize at an equilibrium state conditioned by the input data. Ising machines employ extrinsic mechanisms, such as simulated annealing[57], noise annealing[39], or quantum annealing[7], to guide the spin system towards its ground state rather than local energy minima. These annealing procedures regulate the system's exploration of its energy landscape by adjusting a probability parameter that dictates the system's capacity to escape a given configuration. This probability is controlled by the temperature of the system in simulated annealing, by the noise in the system in noise annealing, and by the tunneling rate between states in quantum annealing. For this demonstration, we utilized the quantum annealing procedure of the D-Wave chip to achieve the free phase of EP (Fig. 1a (free phase) and Fig. 1b ($0 < time < 20\,\mu s$)). The ground state is obtained by progressively reducing the probability from a high value where the system explores

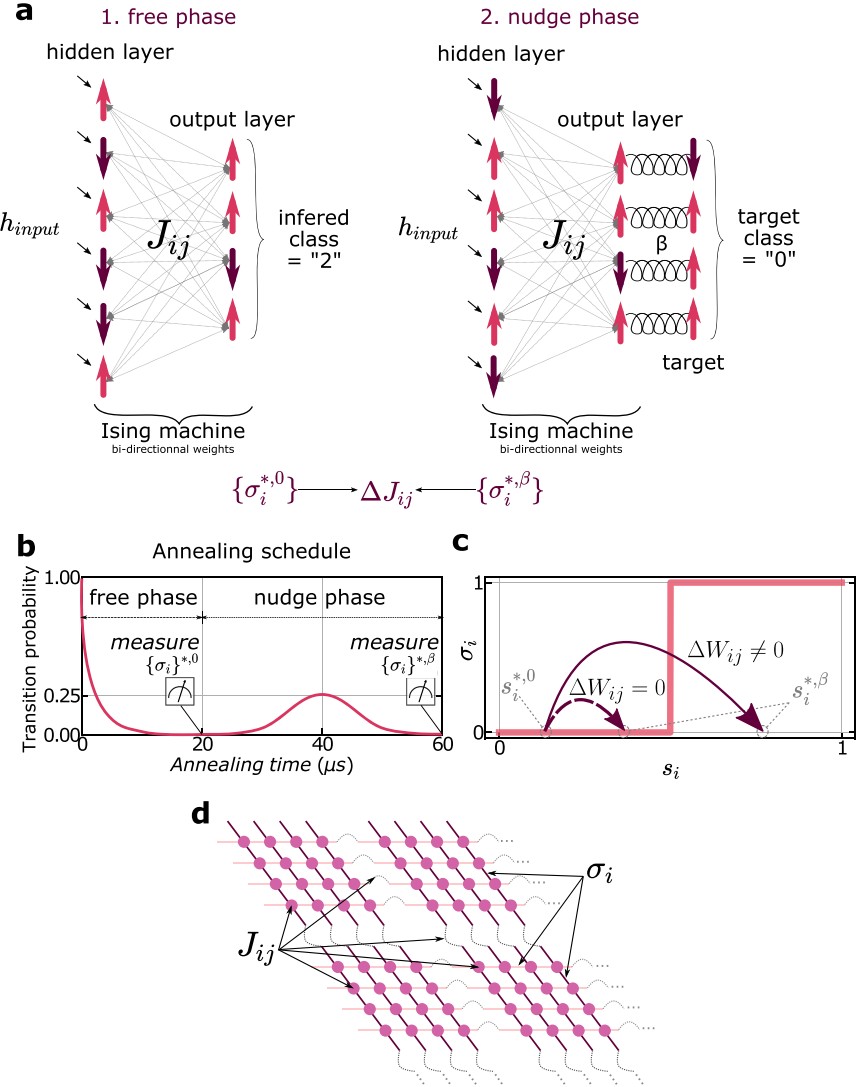

**Fig. 1 | Training the Ising machine with EP. a** Illustration of the free phase and nudge phase of the Equilibrium Propagation algorithm applied to an Ising spin system. For both phases, the input is fed to the chip through bias fields (see Section 1 and 1), with a strength that depends on the task. The steady spins states obtained at equilibrium after the free and the nudge phases can be directly measured on the chip to compute the parameters updates. **b** Annealing schedule used to drive the Ising machine during the two sequential phases of EP. At the end of both phases, the probability of transition between states ends at 0, in the steady state where we measure the states of all the spins. The small bump in the probability during the nudge phase, achieved through reverse annealing, allows the system to be sensitive to the nudge signal applied to the output neurons. **c** Binary activations - such as spins - in a dynamical neural network trained with EP can cause the vanishing gradient issue if the input of the neuron is only weakly modified between the free and the nudge phase. **d** Schematic of the D-Wave chip used with the specific Chimera architecture where spins are arranged as small 4 × 4 fully connected square lattices and laterally coupled to 2 neighbors. The spins $\sigma_i$ are represented as horizontal and vertical lines, whereas the couplings $J_{ij}$ are represented as plain circles for intra-cluster coupling and dotted bridges for cluster-to-cluster coupling.

all possible configurations, towards zero probability (Fig. 1b: $0 < \text{time} < 20\,\mu s$).

The nudge phase of Equilibrium Propagation involves adding to the energy of the system a term proportional to the cost function ($C$) that describes the discrepancy between the output neurons and the target. Here, we use as a cost function the Mean Square Error (see Methods 1) that represents the deviation between the activation functions of the output neurons and their target states:

$$C(\rho(\mathbf{y}),\rho(\hat{\mathbf{y}})) = \sum_{i \in Y} (\rho(y_i) - \rho(\hat{y})_i)^2, \qquad (4)$$

where $Y$ denotes the set of output neurons, the subscript $i$ refers to the index of a specific output neuron and $\rho(\hat{y}_i)$ represents the target value sought for their activation function. For an Ising system, this corresponds to minimizing the spin energy

$$E_{Ising} + \beta \cdot C(\sigma^y,\hat{\sigma}^y) = \sum_{i>j} J_{ij}\sigma_i\sigma_j + \sum_i h_i\sigma_i + \beta\sum_{i \in Y}(\sigma_i - \hat{\sigma}_i)^2, \qquad (5)$$

where the output of the activation function is the binary value of the state $\sigma$. Since the output spin $\sigma^y$ and their corresponding target state $\hat{\sigma}^y$ always take a value equal to $\pm 1$, this formula can be rewritten as (see Supplementary Note 4):

$$E_{Ising} + \beta \cdot C(\sigma^y,\hat{\sigma}^y) = \sum_{i>j} J_{ij}\sigma_i\sigma_j + \sum_{i \notin Y} h_i\sigma_i + \sum_{i \in Y}(h_i - \beta\hat{\sigma}_i)\sigma_i. \qquad (6)$$

This encodes the nudge as an additional bias term $\beta\hat{\sigma}_i$ that is only applied to the output spins.

At the end of the free phase, the system is frozen in its ground state 1b (time = $20\,\mu s$). We found that the application of biases at the output alone is not sufficient to destabilize it, which prevents the error from being backpropagated. To overcome this problem, we employed the Reverse Quantum Annealing procedure[58]. As depicted in Fig. 1b ($20\,\mu s < \text{time} < 40\,\mu s$), we slightly increase the interstate tunneling probability, allowing the system to evolve to a state that is close to the equilibrium state of the free phase but closer to the desired output state. Finally, we decrease the tunneling probability to 0 again, ensuring that the system reaches the new nudged steady state, as illustrated in Fig. 1b ($40\,\mu s < \text{time} < 60\,\mu s$).

While reverse annealing allows for the nudge phase, EP requires additional adaptation to effectively train systems that have abrupt ON/OFF flip-flop activation functions, such as binary neurons or Ising spins. For example, for the same range of input values, a flip-flop neuron is statistically much less likely to change state than a neuron with a continuous activation function when the same nudge bias is applied to them. As illustrated in Fig. 1c, for small nudge biases applied to output spins, the network does not change its state in practice and does not learn. Applying too high nudge biases also poses a problem, as the switching of the output neurons between their two extreme values leads to very strong changes in the states of the neurons of the whole network, hindering the learning process, which needs to be progressive. To solve this problem, Laydevant et al.[56] proposed using several neurons to represent each output class instead of using only one as is commonly done. We have adopted this method, which allows us to induce more flip-flop events in the output and back-propagate them more easily to the rest of the network.

The steady states of the spins at the end of the free and nudge phase (respectively $\sigma^{*,0}$ and $\sigma^{*,\beta}$) are measured, recorded, and used to calculate the gradient of the loss function with respect to the couplings

according to the following learning rule:

$$-\frac{\partial C}{\partial J_{ij}} = \Delta J_{ij} \propto -\frac{1}{\beta}\left[(\sigma_i\sigma_j)^{*,nudge} - (\sigma_i\sigma_j)^{*,free}\right] \qquad (7)$$

for a fully connected architecture. The updates are then applied to the weights using the standard stochastic gradient descent algorithm.

The number of layers, of neurons per layer, and the connectivity of a neural network depend on the task to be solved. Additionally, in our case, this architecture is constrained by the specific connectivity of the hardware system. The D-Wave machine's spins are organized in locally connected sub-networks, as illustrated in Fig. 1d. As a result, mapping a fully connected architecture onto the chip is not straightforward. We employ the embedding procedure provided by D-Wave to map the neural network architecture to the chip's architecture. This procedure relies heavily on the "chaining" process of spins, allowing a spin to couple with more spins than its direct six neighbors. The chaining process involves strongly coupling a chain of physical spins on the chip so that they maintain the same value at each time step of the annealing procedure, effectively implementing a single spin. Through the embedding procedure, we have successfully mapped our fully connected architecture at the expense of using chains of approximately six physical spins per neuron.

## Training a fully connected neural network

We now apply the methods we introduced and described in the previous section to train a neural network featuring a fully connected architecture for recognizing handwritten digits from the MNIST database[59], a widely used benchmark for evaluating the performance of hardware-based neural networks.

Fully connected neural networks for solving the MNIST problem typically consist of several layers of neurons, including an input layer with 784 neurons (one neuron per image pixel), one or more hidden layers, and an output layer. Here we chose to embed the largest possible neural network with one hidden layer as described below. Since Equilibrium Propagation-based training relies on the dynamics of the neurons within the network and input neurons have fixed values, we do not implement them on the chip (see Eq. (8)). Instead, we calculate the product of the input data $X$ (an input image, for instance) and a trainable weight matrix $\mathbf{W_{input}}$ using a digital computer. In our work, $W_{input}$ has the following dimension: $784 \times 120$ (input size × hidden layer size). The resulting vector of fixed bias fields $\mathbf{h_{input}}$ encodes the input on the chip:

$$\mathbf{h_{input}} = \mathbf{X^*W_{input}}. \qquad (8)$$

This bias is then augmented by a second bias that is the equivalent of the standard bias of artificial neural networks: $\mathbf{h_{hidden}} = \mathbf{h_{input}} + \mathbf{h_{bias}}$.

We find that using 4 output neurons per class of digits to be recognized (from 0 to 9, resulting in 40 output neurons) allows the nudge phase to function effectively. When mapping this fully-connected network architecture onto the 5000 locally connected spins of D-Wave, we determined that the maximum number of neurons we can implement in the hidden layer is 120 (Fig. 2a). Due to the limited access time to the D-wave machine, we train this network with only a part of the MNIST data—using 1000 images for training and 100 images for testing (see Methods). We refer to this task as MNIST/100, following the notation of[54]. As shown in Fig. 2b, we obtained a recognition rate of 98.8% (±0.8) on the training data and 88.8% (±1.5) on the test data.

In Fig. 2c–d, we compare the accuracy reached by the physical system to numerical simulations. The first network (dashed lines in Fig. 2c–d) is a spin network identical to the one on the chip, and trained in the same way by replacing the quantum annealing with Simulated Annealing (SA-EP). The second network (solid lines in Fig. 2c) is a

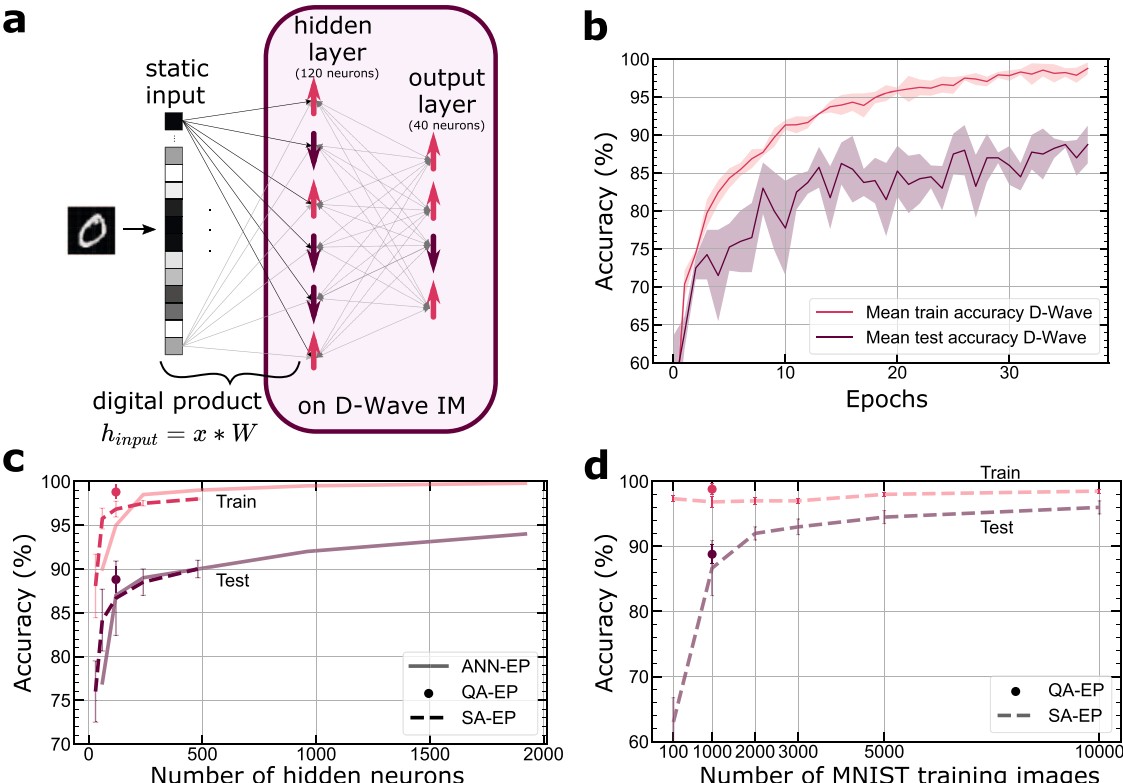

**Fig. 2 | Training a fully-connected neural network on MNIST. a** Embedding the fully connected architecture on the Ising machine. We first compute in software the product between the input vector (an MNIST image) and the first weight matrix. The result is a vector of small constant bias fields that are directly applied to the hidden spins on the chip. The hidden and the output layer are embedded, coupled and eventually nudged on the actual chip. **b** Training on D-Wave: training and testing accuracy as a function of the number of epochs. **c** Training and testing accuracy as a function of the number of hidden neurons. We plot the accuracy obtained with the D-Wave Ising machine (QA-EP) vs. those obtained with Simulated Annealing (SA-EP) and with the deterministic Artificial Neural Network based on binary activations and real-value weights (ANN-EP). **d** Training and testing accuracy as a function of the size of the training dataset. We plot the accuracy obtained on the D-Wave Ising machine (QA-EP) vs. that obtained with Simulated annealing (SA-EP).

software Artificial Neural Network with binary activations and real-value weights (ANN-EP) evolving according to a Hopfield energy and trained by Equilibrium Propagation. The Simulated Annealing along with the Artificial Neural Networks (ANN) are executed on a digital processor as detailed in Methods. Consequently, they establish the benchmark for accuracy that we aim to achieve. As shown in Fig. 2c–d, the accuracy that the hardware spin network reaches on MNIST/100 (denoted by the dots with error bars on 3 repetitions) is always higher or equal to that of the simulated ideal networks for both the training and test databases, demonstrating the quality of the training performed on the Ising machine.

The MNIST/100 task is the most complex task that the D-Wave machine has been trained to solve to date. Previous results, based on unsupervised contrastive divergence learning, have been limited to smaller subsets of the MNIST dataset, such as MNIST/20[10], or images reduced to $(6 \times 6)$ pixels instead of the $28 \times 28$ pixels of the original database[8,12]. The simulations in Fig. 2d indicate that the recognition rate on the test data of the implemented 784-120-40 network can be further improved to 97% ($\pm 1$%) when trained with more images, in this case, 10,000.

The simulations with simulated annealing in Fig. 2c also reveal that the recognition rate on the reduced MNIST/100 database can be further improved to 94% by increasing the number of hidden layer neurons to 1920. While the total number of neurons required for this network in hardware (1960 in total for the two layers, hidden and output) is lower than the total number of spins available on D-Wave, the sparse connectivity of the chip and the need for an embedding step make it impossible to implement in practice. This is because it requires

the use of several spins per neuron, ~6 per neuron for the implemented network (Fig. 2a).

Therefore, it is essential to consider neural network architectures that are congruent with the local connectivity of most Ising machines to make the best use of their resources and, in particular, to use fewer spins per neuron in order to embed larger, and thus more powerful, architectures. In the following section, we demonstrate that it is possible to map and train a complete convolutional neural network on the Chimera graph structure proposed by D-Wave (Fig. 1d), with less than 1.6 spins per neuron on average.

**Training a Convolutional neural network**
In this section, we show that we can directly map a convolutional neural network (CNN) to the Chimera connectivity graph of the D-Wave Ising machine employed in our study.

Convolutional neural networks, which are currently one of the state-of-the-art architectures for image classification, function by sliding convolutional filters over input data (Conv2D in Fig. 3a) in order to extract different learnable features. The resulting feature maps are then down-sampled (Pooling operation in Fig. 3a), combined and fed into a final fully connected classifier (Flattening and Fully connected in Fig. 3a) to assign a class to the input data.

Typically, the convolutional operation is sequential since the filters must move over the input data to compute multiple local dot products (Fig. 3a). However, with the D-Wave Ising machine's Chimera connectivity graph, we can perform this operation in a completely parallel manner by utilizing the local clusters consisting of 4 spins connected to 4 other spins by $4 \times 4$ adjustable couplings represented in Fig. 1d).

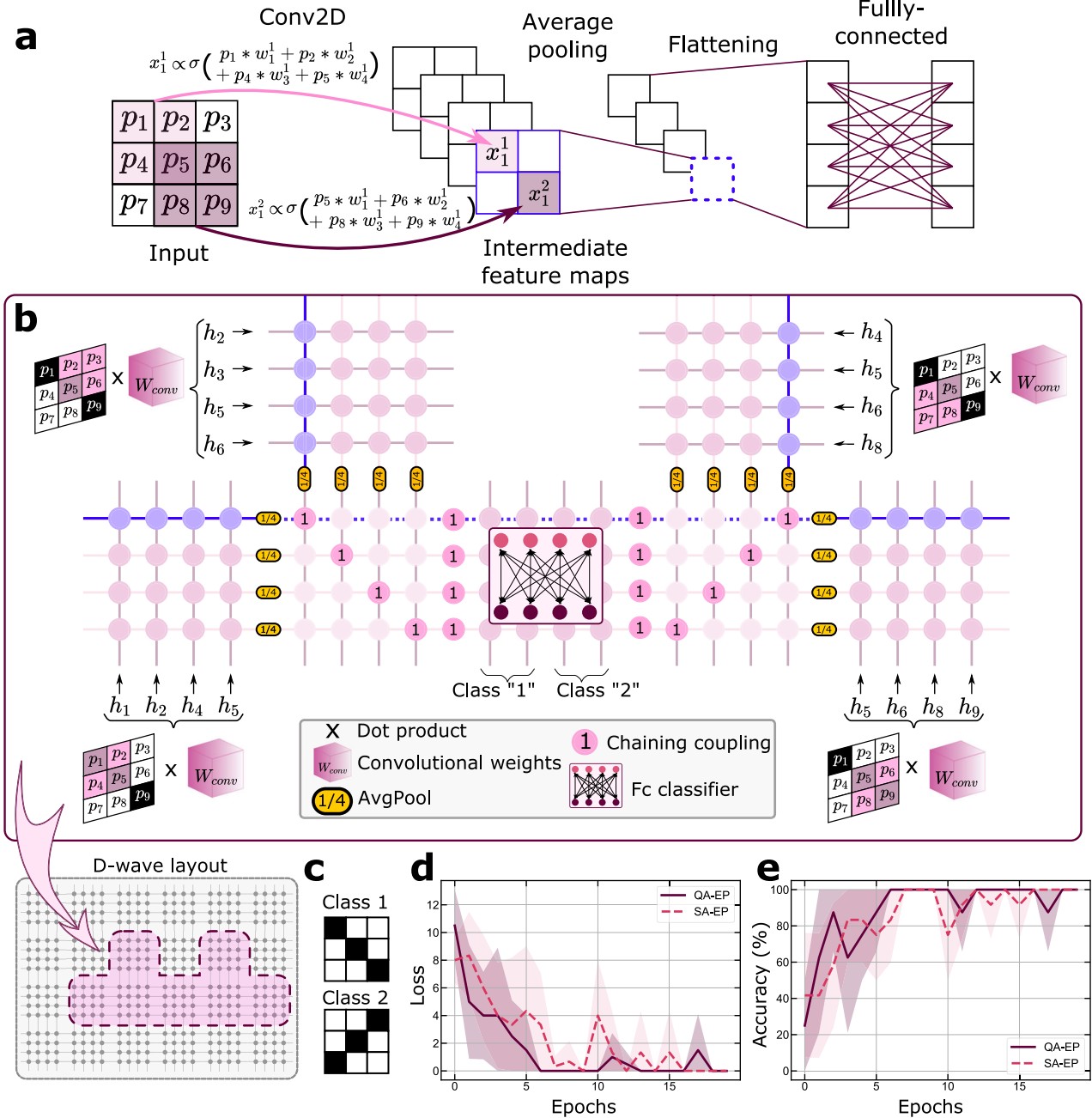

**Fig. 3 | Training a convolutional neural network. a** Detailed view of the convolutional neural network trained on the D-Wave Ising machine. The inputs are $3 \times 3$ pixel images. The convolution layer applies four different sets of $2 \times 2$ weight filters to the input images, generating four $2 \times 2$ feature maps. Each feature map is then condensed to a single value through an average pooling operation. After flattening, a fully connected classifier provides the output of the network, a vector of dimension 4. **b** Schematic of the convolutional neural network's implementation on the Chimera architecture of a D-Wave Ising machine. The four peripheral crossbar arrays perform the convolution operation in parallel. Each array receives a distinct patch of pixels from the input data, yet all share identical couplings, ensuring uniform filter application across different patches. The blue spins on the four crossbar arrays individually represent the values $x_1, x_2, x_3, x_4$ of the feature map highlighted in **a**, obtained by convolving the input with the filter encoded in the couplings depicted in light blue circles. The output of the convolutional operation is then down-sampled via the averaged pooling coupling ($J = \frac{1}{4}$) and linked through identity couplings ($J = 1$). The dotted blue chain represents the output of the averaged pooling operation applied to the feature map depicted with the blue spins. Finally, the results of the averaged pooling operation are fed into the fully connected classifier, which predicts the input's class. Here we have four output neurons as we use two output neurons to encode a class. The convolutional neural network is mapped as-is on the chip, eliminating the need for an embedding step. **c** The training dataset consisting in the 2 patterns used for training the CNN implemented on the D-Wave Ising machine. **d** Training curve (mean squared error) related to training the CNN on the D-Wave Ising machine. **e** Training curve (accuracy (%) related to training the CNN on the D-Wave Ising machine.

The convolutional layer applies a filter, a set of four weights, to extract $2 \times 2$ pixel patches of the image (e.g. $p_1, p_2, p_4, p_5$) and apply a non-linearity (the binary value of the spin), generating a single output value for each patch (e.g., $x_1$). This operation creates four output values for a given filter, constituting a feature map.

We employ four different crossbars, labelled $W_{conv}$ in Fig. 3b to process the four different $2 \times 2$ pixel patches of the input. In the schematic crossbar view, the spins are physically represented by horizontal and vertical lines. The pixel values $p_i$ are applied to the input spins through strong biases $h_i$ that set their direction. The output spins in the

crossbar then align according to the sum of the inputs weighted by the coupling values. The set of couplings highlighted in blue across the crossbars correspond to the same filter, generating the values $x_1, x_2, x_3, x_4$ in the blue output spin of each crossbar. This method allows us to apply four different filters to the full input image simultaneously, thereby executing the entire convolutional operation in parallel.

After the convolution operation, we apply a pooling operation to reduce the dimensionality of the output. Standard CNNs typically employ max pooling, which would, in our example, correspond to keeping the maximum of the four outputs in blue, and coupling only this one to the next layers in the network. We found more convenient and relevant here, given the layout of the chip and the potential degeneracy in the maximum values of a set of binary spins, to implement an average pooling, which averages out the four outputs before connecting them to the next layers. To achieve this, we connect the outputs of the different convolution operations (e.g. the blue "spins" in Fig. 3c) to another chain of spins (e.g. the blue horizontal dotted line in Fig. 3c) through couplings with the value $J = 1/4$, highlighted in yellow in Fig. 3c. This arrangement enables the computation of the weighted average of the convolution outputs and concurrently performs the "flatten" operation of CNNs.

The final stage of the neural network, subsequent to the pooling and flattening operations, consists of a fully connected layer with four outputs. This stage is realized using the central crossbar array depicted in Fig. 3c, where the array's couplings implement the layer's weights.

Here, we use two spins/per class to classify the $3 \times 3$ pixel thumbnails depicted in Fig. 3d. As shown in Fig. 3e–f, the network implemented and trained on the D-Wave's Ising machine achieves a 100% success rate on the training patterns. This result demonstrates the feasibility of training a convolutional network by performing the two phases as well as gradient computations directly on a quantum computer. This method utilizes connectivity far more efficiently than a fully connected network, requiring on average only 1.6 spins per neuron as opposed to approximately 6. These results also demonstrate the power and flexibility of Equilibrium Propagation to train hardware systems with constrained connectivity.

## Discussion

In the past, several attempts have been made to implement neural networks on spin systems by using their physics and taking into account hardware constraints. Boltzmann machines, in particular, can take advantage of the annealing procedures available in Ising machines. However, the size of the networks that can be embedded on the chip is limited in practice because in Boltzmann machines, all the input neurons (784 for MNIST) must be physically present on the chip. Most implementations up to date were made with D-Wave, which offers a few thousand spins. References[8–12] used D-Wave to train a Restricted Boltzmann Machine (RBM) on a coarse-grained version of MNIST down-sampled to $6 \times 6$ pixel images in order to fit the chip that was available at the time of publication. They then fine-tuned the network with back-propagation outside of the chip. Reference[13] only trains some couplings between the hidden nodes (80 nodes) of the Boltzmann Machine. The couplings between the visible and the hidden layer are computed on a side computer. Only[10] trains the Ising Machine on the standard version of MNIST but restricted to 200 training images. However, it is still trained layer-wise and shows limited performances on MNIST/20 (maximum 67% test accuracy with 479 sparsely connected hidden nodes).

In a recent study[14], researchers trained a 2-hidden layer Deep Boltzmann machine with a connectivity similar to the one we used. The simulations were performed digitally, in discrete-time dynamics on an FPGA, contrary to our study that employs the intrinsic dynamics of a physical system to extract the gradients used for training. The Boltzmann machine was kept sparse, avoiding the use of embedding for a fully connected architecture, which is reminiscent of our approach to leverage the Ising machine's connectivity for convolution operations. As we do with Equilibrium Propagation, the authors trained the network as a whole instead of layer-by-layer, which is typically done for Boltzmann machines. They also employed multiple neurons per encoded class (five spins per class, compared to our four spins per class), and showed that their approach converges for systems composed of two-state flip-flop systems, such as Ising machines. Nonetheless, the test accuracy achieved on the full MNIST dataset (90% for both train and test accuracy) underscores the advantage of using Equilibrium Propagation and reverse annealing to train Ising systems. As we showed in the simulations of Fig. 2d, a test accuracy of 97% is theoretically attainable by training the D-Wave Ising machine with Equilibrium propagation on MNIST/1000, a performance that is likely to improve when employing the full database.

In Reference[60], the authors trained a Boltzmann machine on an optical Ising machine, achieving ≈95% accuracy on down-sampled MNIST ($8 \times 8$ pixels). Our approach differs on several points. Their setup uses an FPGA for vector-matrix multiplication and adds a linear classifier on top of a hidden layer, trained separately. We train the full network, including the classifier, directly on the Ising machine using Equilibrium Propagation. Moreover, they use smaller images and more data, sampling their machine 1000 times per problem. In contrast, we sample just 10 times during training and demonstrate that a single sample suffices for post-training inference with binary neurons.

The D-Wave Ising machine has also been employed to train specific components of other neural network types. For instance[61–63], leverage the probabilistic nature of this hardware to generate a sparse latent representation of an auto-encoder. However, again, the authors do not train the entire auto-encoder on the D-Wave Ising machine.

In this work, we demonstrate the feasibility of performing inference, backpropagation of errors, and computation of gradients of a global cost function solely through the dynamics of a spin system. Our approach paves the way for training Ising machines using modern supervised learning algorithms, employing standard gradient-based methods.

Our experiments were carried out using the D-Wave machine, whose quantum properties are a topic of ongoing discussion in the scientific community[64,65]. Our work, however, is fundamentally classical and could be applied to any Ising Machine with the capacity to stabilize in its energy minimum. Although we found that the accuracy obtained on the hardware through quantum annealing is slightly better than software simulations, we cannot conclude on a quantum or classical advantage as the exact training conditions of the hardware cannot be easily replicated in simulations (see Methods). To establish more definitive conclusions on this matter, it would be valuable to investigate whether the D-Wave machine maintains higher accuracy compared to software simulations when trained on other tasks (especially those necessitating an optimal ground state for the Ising system), and ideally, to compare the hardware accuracy with that of classical Ising machines with the same connectivity.

Future generations of D-Wave Ising machines will be capable of modeling larger spin systems, allowing to embed more complex neural networks. In particular, they will be equipped with larger locally connected crossbar arrays[66], enabling the direct application of $3 \times 3$ filters and the performance of convolutions on benchmark images, such as those from MNIST or CIFAR-10. One constraint of our current convolution implementation for progressing in this direction is the requirement for binary inputs. However, there are at least two possible ways to tackle this issue in the future when implementing convolutional neural networks on a D-Wave Ising machine. The first approach is similar to the method we employed in this article for training the fully connected architecture, where we compute the initial vector-matrix product on a digital computer and send the results as inputs to the chip. We could use the same technique with a first convolutional layer computed on a digital computer with real-value inputs, as is commonly done with binary neural networks[67,68]. A second way to manage binary inputs directly on the chip would be to take inspiration from[69], where the

authors stochastically binarize the inputs to accommodate binary inputs, even though the original inputs are real-valued (CIFAR-10).

Finally, Ising machines, designed to reach the ground state of an Ising system, are inherently stochastic in nature[70]. D-Wave for instance operates at finite non-zero temperature, resulting in thermal excitation competing with quantum annealing. When the machine is used for solving combinatorial problems, the state needs to be sampled multiple times to obtain an accurate solution. We also had to sample the state of the machine 10 times at the end of each phase in order to achieve successful trainings. However, we found that after training with our method, the initially stochastic D-Wave Ising machine is brought into a much more deterministic regime (as seen in Fig. 4). This means that after training, the solution is given in a single call to the Ising machine, greatly reducing the inference time. These results also open up the possibility of learning combinatorial problems through a data-driven approach that can provide faster and more accurate solutions than the traditional approach, where Ising system parameters are defined by the problem.

Our results and the algorithm employed to obtain them can be applied to any type of annealing-based Ising machine. Those with an ultra-low power consumption are particularly appealing for reducing the overall electrical consumption of AI and deploying it in embedded systems. Memristors or spintronic nano-components are currently being extensively researched as building blocks for such systems, as they enable the co-integration of memory, novel physical functionality and computing. This greatly enhances the efficiency and scalability of the system, making it more suitable for real-world applications.

The development of unconventional hardware naturally questions the models and the corresponding algorithms to be run on it[71]. For the specific case of hardware or physical neural networks, back-propagation is difficult to realize end-to-end in hardware without major overhead costs (massive peripheral circuitry and memory)[1,28–30].

New learning algorithms grounded in the physics of the hardware are emerging, such as Hamiltonian Echo Backprop[72], Coupled Learning[73], Thermodynamics computing[74,75], Forward-forward-like algorithms[76], Deep reservoir computing[77,78] and Equilibrium Propagation[42]. Hardware demonstrations of those alternative training algorithms are milestones sought after by the unconventional computing community. By physically implementing the spins and couplings, the hardware, which may utilize a variety of technologies such as CMOS, optics or emerging nanotechnologies[1,7,26,28–41], embodies the algorithm with different degrees of abstraction instead of relying on highly synthesized and compiled systems.

In line with this trend, a recent study[45] successfully trained a crossbar array of memristors to emulate the couplings of an Ising-like model using a learning law similar to Equilibrium Propagation. Contrary to our study, the system in this research is not intrinsically dynamic. Instead, the "spin" dynamics is emulated digitally in discrete time by iteratively and recursively interfacing the memristive system to digital electronics.

We show that matching the hardware (a physical system of coupled spins evolving according to the Ising energy - D'Wave system) with the algorithm (a training algorithm harnessing the energy minimization of an Ising energy to find weight updates) is an efficient way to achieve learning in unconventional hardware. Future work could use the methods we have developed here on low power and faster embedded hardware.

In conclusion, this study presents a significant advancement in the field of utilizing Ising machines as hardware platforms for Artificial Intelligence. Leveraging Equilibrium Propagation together with annealing methods, we have successfully demonstrated that Ising machines can be trained using modern supervised learning algorithms, overcoming the limitations of previous attempts to implement neural networks on spin systems. Our experiments, conducted on the D-Wave Ising machine, show that the accuracy obtained through quantum annealing is on par with that of software simulations. Additionally, our results indicate that the local connectivity of Ising machines can be effectively harnessed for performing convolutions. The potential for future developments by combining our approach, where Ising spin systems compute gradients through their intrinsic dynamics, with low-power hardware that employs nanotechnologies such as memristors for implementing local couplings, presents exciting opportunities for the future of embedded AI.

## Methods

In this section, we outline the essential steps of the methods employed to generate the results presented in the paper. Further details can be found in the Supplementary Materials file. We begin by discussing how to interact with the D-Wave Ising machine and identifying the most crucial features to facilitate training with Equilibrium Propagation (EP) on this specific Ising machine. Following this, we provide a brief overview of the requirements and learning rules of EP. Lastly, we delve into the training methods for both the fully-connected and convolutional architectures.

Additionally, we have made our code available at the following link: https://github.com/jlaydevant/Ising-Machine-EqProp to facilitate the reproduction of our results. While access to the D-Wave Ising machine is required for some of the results, similar outcomes can be achieved using Simulated Annealing, which is more easily reproducible. In particular, we provide our modified Simulated Annealing sampler,

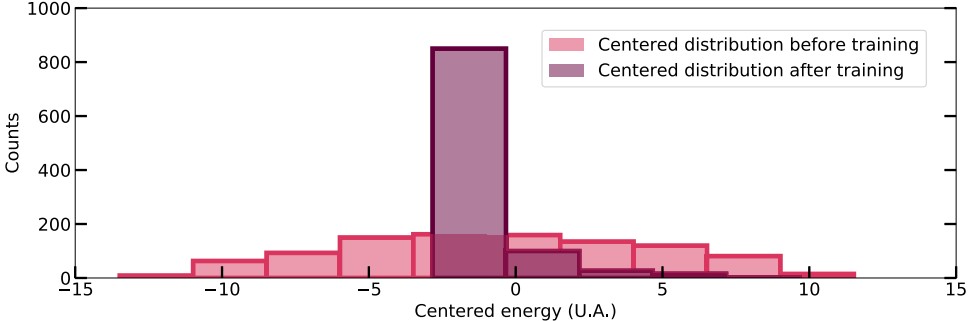

**Fig. 4 | Training the stochastic Ising machine to behave in a deterministic way.** Distribution of the steady-state energy of the spin system modeled by the D-Wave Ising machine when the exact same inputs are applied 1000 times to the same neural network architecture. The light pink distribution corresponds to the energy of the system before training it on MNIST/100. The purple distribution corresponds to the energy of the system after training it with EP. The samples before training have a flat distribution with a large standard deviation - which is the signature of a highly stochastic system. The samples before training exhibit a flat distribution with a substantial standard deviation, indicative of a highly stochastic system. In contrast, the samples after training are concentrated within a narrower region with a considerably smaller standard deviation. This demonstrates that the training procedure causes the Ising machine to exhibit more deterministic behavior.

based on D-Wave's Simulated Annealing code, which we employed to perform both the free and nudge phases with the same annealing schedule baseline, akin to our approach with the D-Wave Ising machine.

The article's text and supplementary material were partially revised by OpenAI's ChatGPT to enhance the clarity and quality of the English.

## Methods for handling the D-Wave Ising machine

The training of neural networks on the D-Wave Ising machine were conducted as follows.

We accessed the Ising machine virtually through a Python API[79] provided by D-Wave. Initially, we needed to define a sampler, which refers to the type of D-Wave machine we intended to use (e.g., Chimera or Pegasus architecture) and the embedding procedure we employed to map the neural network architecture onto the actual architecture of the Ising machine.

Following the initial implementation of EP[42], we performed the two sequential phases of EP on the Ising machine. The input image remained static during all the annealing process, as required for the system to converge given a fixed input. We had to adjust the duration of the annealing for the free phase. We chose the native duration, which is 20 µs (see Fig. 1b). For the reverse annealing we specified the duration and the schedule. We set the duration to 40 µs, the initial annealed fraction to 0, the annealed fraction at time 20 µs to 0.25 and the final annealed fraction to 0 (see Fig. 1b). This enabled the system to change its state according to the nudging signal. This optimal value for the annealed fraction, as determined midway through the reverse annealing process, was first calibrated using a simplified task and then applied to training on the MNIST/100 dataset. Specifically, we employed the same MLP (Multilayer Perceptron) architecture (784-120-40) but trained it on the MNIST/10 dataset. We incrementally adjusted the annealed fraction until the gradients registered on the chip exceeded zero, indicating effective training. This was further validated as the loss diminished and the accuracy exhibited an upward trend.

We want to emphasize that while the quantum annealing procedure is implemented through a dynamical transverse Ising model[80] that allows for quantum fluctuations, the final state obtained through the same quantum annealing procedure is solution of the classical Ising model.

The D-Wave Ising machine is not perfect and thus does not return the ground state of the problem at each call to the solver (see Fig. 4 - distribution before training). This alters the training procedure and we had to sample the problem multiple times per input image to get a good estimate of the ground state. For that purpose, we sampled the same problem 10 times and selected the sample with the lowest energy.

We used the Ising formalism to sample our problems—i.e. the spins are ±1 and not 0/1 as with a QUBO (Quadratic Unconstrained Binary Optimization)—as the D-Wave Ising machine really optimizes the Ising problem. We can submit problems in the QUBO formulation to the D-Wave Ising machine but the parameters (couplings and bias) are scaled non-trivially, which we found to affect the training procedure.

An essential feature to disable when executing a two-phase training algorithm, such as EP, on the D-Wave Ising machine, is the *auto-scale* feature:

$$scaling = \max \left( \begin{array}{c} \max\left(\frac{\max(\mathbf{h},0)}{\max(h_{range},0)}\right), \\ \max\left(\frac{\min(\mathbf{h},0)}{\min(h_{range},0)}\right), \\ \max\left(\frac{\max(\mathbf{J},0)}{\max(J_{range},0)}\right), \\ \max\left(\frac{\min(\mathbf{J},0)}{\min(J_{range},0)}\right) \end{array} \right) \quad (9)$$

This feature enables the scaling (by a factor *scaling*) of problem parameters to optimally fit within the accessible value range for the parameters on the chip, in order to gain in performance. However, when training the Ising machine with EP, we introduce nudging biases on the output neurons (which always have a larger magnitude than the network biases). We immediately see from Eq. (9) that the scaling for all the parameters will change for the second phase. As a result, the optimization problem will be significantly different, making the resulting gradient irrelevant. We found that disabling this feature and manually scaling the parameters (see Supplementary Table 1) allows for the successful training of a neural network on the D-Wave Ising machine. One drawback of this approach is the need to clip the parameters within the available range of values accessible on the chip, which we do after the stochastic gradient descent (SGD) update step.

## Objective function for training a neural network on an Ising machine with Equilibrium Propagation

The particularities of Equilibrium Propagation on Ising spin models have been discussed in the main text. Here, we revisit the choice of the objective function to minimize. In standard software-based simulations of EP, we use the Mean Squared Error (MSE) between the internal state of the output neurons and their target states:

$$C(y,\hat{y}) = \sum_{i \in Y} (y_i - \hat{y}_i)^2 \quad (10)$$

However, when using the Ising machine, we no longer have access to the internal states $y_i$. The only measurement available to us is the activation - the spin's state $\sigma(y_i)$ - of the output neurons, as $y_i$ is implicitly computed through the couplings. This is the reason behind our choice of MSE for training the system (Eq.(4)).

Nevertheless, the choice of using the MSE as the cost function naturally fits the Ising machine frameworks as the minimization of the MSE during the nudge phase simply translates in a nudge bias (see Supplementary Note 4).

## Methods for training a fully connected architecture

We now discuss the two primary challenges associated with training a fully-connected neural network on the D-Wave Ising machine. The first challenge involves embedding a dense graph (the fully-connected architecture) onto a sparsely connected graph, i.e., the architecture of the D-Wave Ising machine. The second challenge concerns efficiently feeding inputs to the Ising machine.

**Embedding the fully connected architecture onto the chip layout.** To map our problem, which is the underlying graph of the neural network we want to train, onto the actual architecture of the chip, we used the *LazyFixedEmbeddingComposite* function. This function operates as follows: during the first annealing, it calls a heuristic function *minorminer*, which aims to find a way to embed the neural network onto the chip layout. However, this procedure is itself an NP-hard problem, resulting in a time-consuming process. *LazyFixedEmbeddingComposite* finds the embedding only for the first annealing and reuses this embedding for subsequent training examples, which significantly reduces the annealing time. In addition to this advantage, it allows the training process to account for the local imperfections of the chip (faulty spins, noise, etc.) so that the parameters are updated based on the actual dynamics of the chip.

We want to emphasize that the graph embedding procedure is performed only once at the beginning of the training, and this specific embedding is consistently used throughout the training duration. This means that the potentially resource-intensive graph embedding step gets spread out, making it more cost-effective and amortized over numerous training iterations. As highlighted in the section discussing the convolutional neural network (Section 1), we anticipate that future

architectures driven by layout (which are entirely independent of the graph embedding procedure) will offer greater scalability.

The embedding process relies heavily on the chaining procedure - using multiple hardware spins to represent a single spin in order to couple it to more neighbors than the architecture allows for - and the chaining strength can be adjusted. In practice, we set the chaining strength (i.e., the value of the couplings between the hardware spins that represent the same spin) to −1 which has proven effective. Moreover, spins within a single chain may not all align post-annealing, leading to 'chain breaking.' We address this through a straightforward majority-vote strategy. While chain breaking poses significant challenges in contexts requiring highly precise ground states, such as combinatorial optimization, Equilibrium Propagation (EP) merely necessitates reaching a stable equilibrium state[42]. As a result, we expect neural network training via EP on Ising Machines to be far more tolerant of chain-breaking complications compared to combinatorial problem-solving.

For training a fully-connected architecture, we used the Pegasus architecture on the *Advantage 4* chip available at the time of the simulations.

**Feeding the input data to the chip.** As mentioned earlier, the input image remains static throughout the entire annealing process, as required for EP. This enables us to perform the vector-matrix product between the input image and the first weight matrix in silico only once, and apply the result as an input bias to the hidden neurons. This approach allows us to train the Ising machine with a large input (MNIST: $28 \times 28$ pixels), which was previously impossible with existing training methods that required embedding the input data on the chip[8,14]. This weight matrix is also trained, given only the steady-state spin states in the Ising machine and the input data.

**Training data.** Training a neural network reduces to an iterative process where the training data is shown multiple times to the network. So training a neural network on the D-Wave Ising machine requires multiple calls to the solver, each being billed according to the time the solver is used.

For instance, a free phase is billed the following way: time to initialize the parameters on the chip (couplings and field biases): $\approx 7$ $ms$, time to thermalize the chip after the initialization of the parameters: $\approx 1\,ms$, annealing time: $\approx 20\,\mu s$ being done 10 times per example in our case, readout and thermalization time before next annealing: $\approx 200\,\mu s$ which result in $\approx 10\,ms$ per free phase. The nudge phase is even more costly in time as we need to re-initialize the spins for each reverse annealing which adds 10*7 $ms$ per example and gives $\approx 80\,ms$. In total this results in $\approx 9\,ms$ per training example (free and nudge phases). An epoch on full MNIST (60k training images and 10k testing images) would be $\approx 5525\,s$ on the solver. And 50 epochs would take $\approx 5k$ min which is $\approx 77\,h$. This would result in an insane bill, which is why we chose to use a subset of MNIST: MNIST/100. An epoch with 1000 training examples would be $\approx 91\,s$ on the solver, 50 epochs would take $\approx 76\,min$ which is $\approx 1.27\,h$. So to train one neural network on the D-Wave Ising machine for 50 epochs on MNIST/100 we have to pay for $\approx 1.27\,h$ access time, which is much more accessible than the 76$h$ for full MNIST.

Following the notation of [54], our dataset is called MNIST/100, which contains 1000 training images with 100 training images per class, and 100 testing images with 10 testing images per class. To create our training (resp. testing) dataset, we select the first 100 (resp. 10) images of each class from the MNIST dataset downloaded with Pytorch. This equidistribution helps avoid training (resp. testing) bias in such a small training (resp. testing) dataset. We did not use data augmentation techniques.

Furthermore, to save access time to the Ising machine, we adopted a simple scheme similar to refs. 22,81. At the end of the free phase,

we compared the state of the output layer of the network to the target. If the output layer was equal to the target vector, we skipped the nudge phase for that particular input data, meaning no gradient had to be computed for this image. This approach significantly saved access time and accelerated the training.

**Learning rule for the fully connected architecture.** The D-Wave Ising machine minimizes the following Ising energy function:

$$E_{Ising} = \sum_{i>j} J_{ij}\sigma_i\sigma_j + \sum_i h_i\sigma_i \tag{11}$$

Thus, the learning rule for the couplings is directly derived from Eq. (11):

$$\frac{\partial \mathcal{L}}{\partial J_{ij}} = \frac{1}{\beta}\left(\frac{\partial E_{Ising}}{\partial J_{ij}}(x,\theta,\sigma^{*,\beta},\beta,\hat{\sigma}) - \frac{\partial E_{Ising}}{\partial J_{ij}}(x,\theta,\sigma^{*,0},0,0)\right) \tag{12}$$

where $\sigma^{*,0}$ and $\sigma^{*,\beta}$ stands for the two sequential free and nudge equilibrium states.

Finally, Eq. (12) directly read as:

$$-\frac{\partial \mathcal{L}}{\partial J_{ij}} = \Delta J_{ij} = -\frac{1}{\beta}\left((\sigma_i\sigma_j)^{*,\beta} - (\sigma_i\sigma_j)^{*,0}\right) \tag{13}$$

where $\mathcal{L}$ stands for the loss function to be minimized during the training procedure - here the Mean Squared Error function.

This learning rule differs from a negative sign to the conventional learning rule of fully-connected layers with EP as conversely to the standard energy minimized in EP, the D-Wave Ising machine minimizes this specific Ising energy function (Eq. (11)) where there is a positive sign in front of the coupling sum.

Similarly, we can derive the learning rule for the bias fields:

$$-\frac{\partial \mathcal{L}}{\partial h_i} = \Delta h = -\frac{1}{\beta}\left(\sigma_i^{*,\beta} - \sigma_i^{*,0}\right) \tag{14}$$

We feed these gradients to a SGD optimizer without momentum and no mini-batch:

$$J \leftarrow J + \eta \cdot \Delta J \tag{15}$$

where $\eta$ stands for the learning rate which is a tunable parameter.

## Methods for training a convolutional architecture
In this section, the main challenge, besides training the system with EP, is to find the correct embedding that implements the convolutional neural network we want to train.

**Handcrafting the embedding dictionary.** Contrary to the embedding procedure for the fully-connected architecture, we handcrafted the embedding for this case, as we leveraged the actual architecture to do specific computations. To achieve this, we manually created a dictionary that maps the index of the spins in our architecture to a specific site on the chip. For the spins requiring chaining (e.g., for implementing the results of the average pooling operation), the dictionary maps the spin to a list of hardware sites.

For the convolutional architecture, we used the Chimera architecture on the chip *DW-2000*.

A typical embedding dictionary is:

```
embedding = {0 : [560], 1 : [561], ..., 39 : [579]}
```

where the key of the dictionary is the index of a specific neuron in the architecture and the linked list is the corresponding spin(s) on the chip. The embedding is shown in Supplementary Fig. 12.

**Feeding the input data to the chip**. The concept behind the convolutional architecture is centered around the "local" product of input data and the coupling. To demonstrate that this works in practice for realizing convolutional operations, we had to embed the inputs directly on the chip, unlike the fully-connected architecture. For that purpose, we set strong biases (i.e., ±4, the largest possible value for biases on DW-2000) on the spins corresponding to the inputs. This ensures that they remain constant throughout the entire annealing procedure and have the binary value corresponding to the sign of the bias applied to them.

**Learning rule for the convolutional architecture**. The learning process for the convolutional architecture differs from that of the fully-connected model. The operations involved in the energy function have changed, resulting in a different learning rule for the convolutional weights. However, the learning for the last weights involved in the classifier remains the same as for the standard fully-connected architecture.

The learning rule for the convolutional weights **J** between the input $x$ and the "hidden" layer $h$ - after activation but before the pooling operation - is:

$$\Delta \mathbf{J} = \frac{1}{\beta} \left[ (x \star h)^{*,\beta} - (x \star h)^{*,0} \right] \qquad (16)$$

where $\star$ denotes the convolution operation between $x$ and $h$. Here the gradient $\Delta \mathbf{J}$ has the same dimension as the convolutional weights tensor **J**, so we can directly apply the update through stochastic gradient descent:

$$\mathbf{J} \leftarrow \mathbf{J} + \eta \cdot \Delta \mathbf{J} \qquad (17)$$

The learning rule Eq. (16) can be easily extended to the layers that are not directly linked to the fixed inputs:

$$\Delta \mathbf{J} = \frac{1}{\beta} \left[ \left( h^{\ell} \star h^{\ell+1} \right)^{*,\beta} - \left( h^{\ell} \star h^{\ell+1} \right)^{*,0} \right] \qquad (18)$$

where $h^{\ell}$ and $h^{\ell+1}$ stands for the states of the two consecutive layers $\ell$ and $\ell+1$.

**Methods for the simulated annealing simulations**
We performed digital simulations using Simulated Annealing (SA) to benchmark the results obtained on the D-Wave Ising machine. SA shares the same goal as quantum annealing algorithms, but relies on temperature to control the probability of a system to escape from a given configuration. The temperature is initially set to a high value, allowing the system to explore various configurations. It is then gradually decreased, so that y the end of the annealing process, the system reaches a steady state (see Alg. 2).

We used a code provided by D-Wave to perform those simulations.

However, similarly to the "auto-scale" feature that had to be disabled on the D-Wave Ising machine, we manually set the temperature range, which is natively auto-scaled to match the range of parameters for the problem to be solved through simulated annealing. The temperature range is chosen based on the initial parameter values and remains fixed throughout the training. Although the temperature range may not be ideal, it works well in practice.

We also modified the original code to implement a similar kind of reverse annealing to the one of D-wave, but with temperature. In this case, we reversed the temperature schedule during the first annealing for a certain duration up to a tunable value (as with simulations on D-Wave) and then decreased it back to zero. The temperature schedule - i.e., the schedule of $p_{1\rightarrow 2}$ as here $p_{1\rightarrow 2} \propto e^{-\frac{\Delta E_{1\rightarrow 2}}{T}}$ - follows the same curve as in Fig. 1b.

To run the simulations, we integrated the Simulated Annealing sampler into the code developed for the training on the D-Wave Ising machine. The user simply needs to specify which sampler to use for a specific training.

Similar to the trainings on the D-Wave Ising machine, the states reached with SA are stochastic. As a result, we also had to sample the states multiple times per image to obtain a reliable estimate of the ground state. Additionally, we skipped the nudge phase when the free phase ad already produced the correct output state.

**Methods for the deterministic simulations**
We also conducted digital simulations using a similar type of neural network: one with binary activations and real-valued weights but with deterministic dynamics, i.e., a gradient dynamics on the energy function.

For these simulations, we adapted the code from[56], in which the neurons are described by the standard EP energy function:

$$E_{EP} = \sum_i s_i^2 - \sum_{i>j} W_{ij} \rho(s_i)\rho(s_j) - \sum_i b_i \rho(s_i) \qquad (19)$$

where the weights $W_{ij}$ are real-valued and $\rho$ is the binary Heaviside step function (see Fig. 1c):

$$\rho(s) = \begin{cases} 0 \; if \; s < 0 \\ 1 \; else \end{cases} \qquad (20)$$

A dynamical equation for the neurons that minimizes this energy function is given by the following gradient dynamics:

$$\frac{ds}{dt} = -\frac{\partial E_{EP}}{\partial s} \qquad (21)$$

where $s$ stands for the internal state of a specific neuron in the network. For a particular neuron $s_i$, we can simply rewrite this equation as:

$$\frac{ds_i}{dt} = -s_i + \rho'(s_i)\left[ \sum_j W_{ij}\rho(s_j) + b_i \right] \qquad (22)$$

This equation governs the internal dynamics of the binary neurons - with the particular choice of $\rho'(s_i) = 1_{0<s_i<1}$ which is arbitrary as the derivative of the Heaviside step function is almost zero everywhere.

We use an Euler scheme to solve this dynamics:

$$s_{t+dt} = s_t + dt^* \frac{ds}{dt} \qquad (23)$$

where the time step $dt$, the number of time steps for the free and nudge phases resp. $T$ and $K$ are hyperparameters to tune (see Supplementary Table 3). No annealing is used here so given an initial state (always 1 here) and a set of parameters, the dynamics always converges toward the same steady state so we do not need to repeat the simulation for each image.

For the nudge phase, the energy function is augmented by a cost term as follows:

$$\frac{ds}{dt} = -\frac{\partial E_{EP}}{\partial s} - \beta \frac{\partial C}{\partial s} \qquad (24)$$

where $C$ is the cost function to be minimized and $\beta$ the nudging parameter.

The output neurons $y$ now have the following dynamics:

$$\frac{dy_i}{dt} = -y_i + \rho'(y_i)\left[\sum_j W_{ij}\rho(s_j) + b_i\right] + \beta \cdot (y_i - \hat{y}_i) \tag{25}$$

where $\hat{y}_i$ is the target state for the corresponding neuron $y_i$.

We compute the gradients given the free and nudge equilibrium states:

$$-\frac{\partial \mathcal{L}}{\partial W_{ij}} = \Delta W = \frac{1}{\beta}\left((\rho(s_i)\rho(s_j))^{*,\beta} - (\rho(s_i)\rho(s_j))^{*,0}\right) \tag{26}$$

and feed them to a SGD optimizer (Eq. (15)).

## Data availability
The datasets analyzed, and all data measured in this study are available at: https://doi.org/10.5281/zenodo.10690111.

## Code availability
The code to reproduce the results is available on github at the following link: https://github.com/jlaydevant/Ising-Machine-EqProp.

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

## Acknowledgements

This work was supported by the European Research Council advanced grant GrenaDyn (reference: 101020684). The text of the article was partially edited by a large language model (OpenAI ChatGPT). The authors would like to thank D. Querlioz for discussion and invaluable feedback.

## Author contributions

J.G., J.L. and D.M. devised the study. J.L. performed all the simulations and experiments. J.G and J.L. wrote the initial version of the manuscript. All authors discussed the results and reviewed the manuscript.

## Competing interests

The authors declare no competing interests.
