## [Peer Review File · Nature Communications]

REVIEWER COMMENTS

Reviewer #1 (Remarks to the Author):

The authors have applied equilibrium propagation (EP) toward the supervised training of an Ising machine and implemented the work experimentally in a D-Wave quantum annealer. MNIST classification and convolutional layers were shown, combining experiments and some digital computer processing.

Overall, I find the work interesting and thought-provoking. The scientific approach is well-grounded and can benefit practitioners in this field because the authors devised several new techniques to adapt their EP to the hardware constraints in their system. In this way, the experimental demonstrations are valuable, even if quite limited in size and scale as the authors acknowledged.

At a high level, I had three significant puzzles I would ask the authors to help explain in the paper.

1) Is not the network model that is considered here simply a binary neural network, as has been explored for quite some time in the artificial neural network community to reduce memory/hardware requirements [e.g., Lin, et. Al, "Towards accurate binary convolutional neural network." NeurIPS(2017)]? It feels like a more constrained binary neural network trained with equilibrium propagation. Is this, then a more constrained network than what was studied in References 24 and 33?

2) What is the significance of the D-Wave quantum annealer in this work? It seems that no quantum properties were needed or used for the present work.

3) As a follow-up to 2), if the quantum annealing is not playing a key role here, then I would ask the authors to elucidate what are the key differences from Ref 31, which seemed to similarly involve training of neural networks combining EP and an Ising Machine or Hopfield network (which appear identical). Ref 31 used a purely classical annealer to find the energy minima, followed by EP, while the present work appeared to use a quantum annealer to find energy minima, followed by EP. Is this an incorrect understanding? In a related way, it would help to distinguish the present work from Ref 14, which also seems to have similarities.

Some additional specific questions and comments:

- In Eqn 8, it would be helpful to specify the exact matrix dimensions used for W_{input} in this study
- Related to Figure 2c and 2d, a comparison was made to SA and ANN-EP. For “SA” it was not clear if SA was only to dynamically evolve the activations (spins/neurons). Was EP still used in this case to update the weights J_{ij} , or was a different procedure used than Eqn 7?
- Minor observation: in Supplemental “Algorithm 3”, step 7, why is there a term computed that is multiplied by β , if β should be 0? Probably just a bug.
- Minor typo: Page 8, top, “as illustrated in n Fig. 1d.”
- Minor typo: Page 12, halfway, a sentence starts with “his stage is realized”

Reviewer #2 (Remarks to the Author):

This paper proposes an approach to train Ising machines in a supervised way. The approach is based on the Equilibrium Propagation method, which the authors apply to the DWAVE machine. I enjoyed reading the paper. It is very well written, the results novel and sound. The paper opens up a new application domain for Ising machines beyond combinatorial optimisation. Therefore, I believe the results are interesting to a broad community both for machine learning specialists as well as the large numbers of teams that are now building Ising machines. I also specifically liked the very careful assessment of the possibility of a quantum advantage.

Ising machines have been influential in the very beginnings of AI in the unsupervised training of generative networks, especially in the framework of Boltzmann machines. The authors touch upon this in their introduction. Although, it’s not clear if Boltzmann machines underperform due to the difficulty of correctly sampling the Boltzmann distribution and therefore having to rely on approximate methods for this. I think this is an important point to highlight better as the authors compare their results to unsupervised contrastive divergence learning on DWAVE.

The authors refer to Ising machines as binary spin systems. A lot of the properties that the authors refer to are also related to binary spin systems. However, the umbrella term of Ising machines encompasses much more. There are classical systems such as from the Rome group (Pierangeli2019). There is also a large variety of analogue Ising machines implementing the spins not as binary values but as analogue values e.g. Ising machines developed by Stanford (McMahon2016), NTT (Inagaki,2016), and many more. No mention is made of these works in the paper, while the authors do refer to the distinct advantages of the DWAVE. It is therefore unclear to what they compare. As an example the authors write that most

Ising machines have local connectivity. This is true for DWAVE, but the systems mentioned above can have full connectivity. In the same context, the Brussels group have shown (in Bohm2021) that Ising machines can be used for unsupervised learning on the MNIST and it would be interesting to see a comparison to this work.

In the description of the algorithm, there is a reliance on the tendency of Ising machines to reach the ground state. However, for many graphs, Ising machines tend to get stuck in local minima instead of the ground state. How big of an issue would this be?

Finally, the discussion of the Ising machine reaching a more deterministic regime (in Fig 4): this is something I'm quite unsure of how one would interpret this. Isn't this more likely related to the task that is being trained. I.e. a task with a more simple energy landscape where the DWAVE always finds the ground state and by this removing the stochastics. Maybe, the authors can expand on this discussion. e.g. would this carry over to other tasks?

Minor comments:

- the spreads in Fig 2c are not very visible
- the text contains some typos and some references are incomplete
- the labels of fig 3a labels are very small

Reviewer #3 (Remarks to the Author):

In "Training an Ising Machine with Equilibrium Propagation", the authors use a commercially available D-Wave quantum annealer to perform a rudimentary learning task through the use of a much smaller version of the celebrated MNIST dataset. The authors use the Equilibrium Propagation (EP) algorithm.

Despite enjoying the originality of the manuscript as an impressive hardware experiment (where a good portion of the algorithm is performed on a digital computer), I have fundamental concerns about the eventual significance of such a study. Below are my general concerns:

[1] EP is a physics-inspired learning algorithm that has now been around for 7 years and despite the authors' enthusiasm (based on references to mostly their own papers), it is unclear whether EP really delivers in reaching state-of-the-art performance in modern ML. The literature review the authors perform largely focuses on critiquing Boltzmann Machines, another algorithm, whose relevance is questionable, as the authors are well-aware. The authors might want to show independent evidence as to whether EP can match the performance of the actual state of the art in ML, which is a moving and tough target to match.

[2] Second issue is the extremely scarce experimental data in the present version. One excuse given is "limited access time" to the D-Wave machine, whose meaning is vague in a scientific context.

Moreover, the "fully connected" network results in Figure 2 contain only a few data points on an extremely toy problem. Crucial details about the minor graph embedding (MGE) procedure, the well-known "chain-breaking" between ferromagnetic spins are absent. Chain-breaking often changes the "actual" ground state of an embedded problem which could affect the sensitive nature of relaxation in the EP algorithm. It is unclear if the authors are aware of this problem or paid attention to it because it is unaddressed. Additionally, I have found the critical discussion regarding how the qubits are "stuck" in the nudging phase of the algorithm lacking. It sounds like there was a very delicate fine-tuning to get those few data points which would be exacerbated in real problems.

[3] Third is the intentional or unintentional conflating of "Ising Machines" with the "D-Wave Machine". Most importantly, the D-Wave machine does NOT implement the classical Ising model but rather the transverse field Ising model (unfortunately not even mentioned even superficially). This difference may sound unimportant, however, the dynamics of the TFIM and its equivalence to the Ising model have been a topic of intense discussion.

The remarks about why the D-Wave Machine is the best available Ising Machine are simply not correct: despite its possible advantages that are only recently becoming clear [see <https://www.nature.com/articles/s41586-023-05867-2>], D-Wave is among the worst performing Ising Machines in optimization (see, Matthew Kowalsky et al 2022 Quantum Sci. T) due to the aforementioned chain-breaking, short coherence times and low bit precision. Same issues persist in sampling problems as well.

There are many other, larger Ising Machines that could be used for an EP-training on "Ising machines". That the authors laud the bit precision of D-Wave while casting their study in the context of Ising

Machines is quite surprising to put it mildly: I strongly recommend they change their title or actually train their algorithm using an IM, wherever this paper is published.

Please note that this is not a criticism of the D-Wave machine (which has already shown better scaling in select problems) but rather the strategic focus of the present manuscript on something it does not address.

[4] The final problem I feel unaddressed is the following: why should we go through the difficult procedure of minor-graph embedding, securing a D-Wave annealer, and having a hybrid classical computer to perform the gradient descent and input multiplication, as done by the authors? Why would a software simulation of simulated annealing (as the authors have done in Figure 2) not be sufficient for whatever purpose we have? The authors seem to be talking about "low power" and "intrinsic dynamics" but the question is really not answered. The authors' claim about how their D-Wave implementation "beats simulated annealing" in every task without a clear discussion of hyper-parameters is very flimsy. I strongly doubt their D-Wave training method can withstand a properly optimized SA, and this is hard to tell with the scant details in the paper.

Response to reviewers

We thank the reviewers for their questions and comments. Following their suggestions, we have clarified the manuscript, expanding on the motivations for using the D-Wave machine for this demonstration, the classical nature of our study, the details of the model and the perspectives of the work towards low power AI-dedicated hardware.

Reviewer #1 (Remarks to the Author):

The authors have applied equilibrium propagation (EP) toward the supervised training of an Ising machine and implemented the work experimentally in a D-Wave quantum annealer. MNIST classification and convolutional layers were shown, combining experiments and some digital computer processing.

Overall, I find the work interesting and thought-provoking. The scientific approach is well-grounded and can benefit practitioners in this field because the authors devised several new techniques to adapt their EP to the hardware constraints in their system. In this way, the experimental demonstrations are valuable, even if quite limited in size and scale as the authors acknowledged.

At a high level, I had three significant puzzles I would ask the authors to help explain in the paper.

1) Is not the network model that is considered here simply a binary neural network, as has been explored for quite some time in the artificial neural network community to reduce memory/hardware requirements [e.g., Lin, et. Al, "Towards accurate binary convolutional neural network." NeurIPS(2017)]? It feels like a more constrained binary neural network trained with equilibrium propagation. Is this, then, a more constrained network than what was studied in References 24 and 33?

Our work is indeed based on a neural network model that is more constrained than the one of References 24 and 33. The latter references rely on digital simulations where both neural activations and parameters (weights and biases) trained through Equilibrium maintain full precision.

Contrastingly, our neural network model employs binary activations (the physical spin values). However, the analogs to synaptic couplings (the interconnections between spins) are not strictly binary. Specifically, the D-Wave hardware we utilized has a precision of 5-6 bits. Thus, our model bridges the gap between binary neural networks, where both activations and synaptic weights are binary, and conventional neural networks, in which neither activations nor synapses are binary.

We have modified the end of the introduction to clarify this point:

“We train the Ising machine with Equilibrium Propagation, taking advantage of its capacity to reach the ground state of its energy function through annealing during the free phase and reverse-annealing during the nudge phase. While the coupling between the spins of the D-Wave machine has a high precision (5-6 bits) approaching the full precision synapses of the

original Equilibrium Propagation model \cite{kendall_training_2020, laborieux_scaling_2020}, the spins of the machine correspond to neurons with binary activations. In order to train this binary system, we adapt procedures developed for training binary neural networks \cite{Laydevant_2021_CVPR, Lin} and increase the number of outputs neurons.”

2) What is the significance of the D-Wave quantum annealer in this work? It seems that no quantum properties were needed or used for the present work.

Indeed, our work, while performed on a quantum computer, is fundamentally classical. Although we utilized the D-Wave machine in our experiments for its efficient quantum annealing property and its convenience as the publicly available Ising machine with the largest number of spins and directly reconfigurable coupling parameters, our primary focus is not to advocate for a quantum advantage, but to demonstrate training results that could be applicable to any Ising machine.

We have modified the text of the discussion to clarify this point:

“Our experiments were carried out using the D-Wave machine, whose quantum properties are a topic of ongoing discussion in the scientific community [48, 49]. Our work, however, is fundamentally classical and could be applied to any Ising Machine with the capacity to stabilize in its energy minimum. Although we found that the accuracy obtained on the hardware through quantum annealing is slightly better than software simulations, we cannot conclude on a quantum advantage as the exact training conditions of the hardware cannot be easily replicated in simulations (see Methods). To establish more definitive conclusions on this matter, it would be valuable to investigate whether the D-Wave machine maintains higher accuracy compared to software simulations when trained on other tasks (especially those necessitating an optimal ground state for the Ising system), and ideally, to compare the hardware accuracy with that of classical Ising machines with the same connectivity.”

3) As a follow-up to 2), if the quantum annealing is not playing a key role here, then I would ask the authors to elucidate what are the key differences from Ref 31, which seemed to similarly involve training of neural networks combining EP and an Ising Machine or Hopfield network (which appear identical). Ref 31 used a purely classical annealer to find the energy minima, followed by EP, while the present work appeared to use a quantum annealer to find energy minima, followed by EP. Is this an incorrect understanding? In a related way, it would help to distinguish the present work from Ref 14, which also seems to have similarities.

1. In Reference 31, the vector-matrix multiplication between the states of the spins at time t and the couplings values is done in the analog domain with a memristive crossbar array, but the dynamics of the spins is emulated digitally and requires 1 ADC, 1 DAC and 1 comparator per spin to emulate the discrete-time dynamics (Fig. 14 SM of Reference 31). Therefore, the computing is performed on a system that is not intrinsically dynamical and does not naturally evolve to minimize its energy. We are, on the other hand, computing with a physical system of spins that naturally minimize their energy following the spirit of the original Equilibrium Propagation algorithm.

We have modified the discussion to clarify the comparison: “In line with this trend, a recent study \cite{Yi2022} successfully trained a crossbar array of memristors to emulate the couplings of an Ising-like model using a learning law similar to Equilibrium Propagation. Contrary to our study, the system in this research is not intrinsically dynamic. Instead, the "spin" dynamics is emulated digitally in discrete time by iteratively and recursively interfacing the memristive system to digital electronics. Combining our

approach, in which an Ising spin system computes gradients through its intrinsic dynamical evolution towards a minimum of energy, with the use of memristive couplings presents significant potential for future low-power, embedded Ising machines that can be trained on the edge.”

2. In Reference 14, as in ref. 31, the authors do not use a physical system that naturally evolves to an energy minimum to compute: they simulate the equations digitally in discrete time with an FPGA. In addition, they train a Boltzmann machine which, while based on an energy-based neural network such as the models trained with Equilibrium Propagation, is a generative model. In our work (and in the work of Reference 31) we train a discriminative model with Equilibrium Propagation which is the state of the art for classification tasks. The similarity between Reference 14 and our work is the sparse connectivity of the D-Wave hardware they use as the connectivity graph for their Boltzmann machine.

We have modified the paragraph of the conclusion in which we compare our work to Ref. 14 as follows: “In a recent study \cite{niasi2023training}, researchers trained a 2-hidden layer Deep Boltzmann machine with a connectivity similar to the one we used. The simulations were performed digitally, in discrete-time dynamics on an FPGA, contrary to our study that employs the intrinsic dynamics of a physical system to extract the gradients used for training. ...”

Some additional specific questions and comments:

- In Eqn 8, it would be helpful to specify the exact matrix dimensions used for W_{input} in this study

We added the exact matrix dimension in the manuscript so it will be clearer to the reader that that this matrix has the dimension $\text{input_size} \times \text{First_hidden_layer_size}$ so 784×120 in our case.

- Related to Figure 2c and 2d, a comparison was made to SA and ANN-EP. For “SA” it was not clear if SA was only to dynamically evolve the activations (spins/neurons). Was EP still used in this case to update the weights J_{ij} , or was a different procedure used than Eqn 7?

Thank you for pointing out this source of misunderstanding. Indeed, we have always used EP as the training algorithm, independently of the dynamics implemented: QA (Quantum annealing - on the hardware), SA (Simulated annealing - digital simulation) or ANN-EP (Emulated deterministic gradient-based dynamics - digital simulations). All dynamics ensure the convergence of the system to an equilibrium state as required for EP to work (both through annealing: QA and SA, or the gradient-based dynamics: ANN-EP).

We have changed the labels to QA-EP, SA-EP, ANN-EP in the text to clarify this point.

- Minor observation: in Supplemental “Algorithm 3”, step 7, why is there a term computed that is multiplied by β , if β should be 0? Probably just a bug.

Thank you for pointing out this error. Effectively the symbol “ \neq ” should be the symbol “ $=$ ”. We modified “ \neq ” to “ $=$ ” in the Supplementary materials.

- Minor typo: Page 8, top, “as illustrated in n Fig. 1d.”

Thank you, we deleted the extra “n”.

- Minor typo: Page 12, halfway, a sentence starts with “his stage is realized”

Thank you, we added the missing “T” at the beginning of the sentence.

Reviewer #2 (Remarks to the Author):

This paper proposes an approach to train Ising machines in a supervised way. The approach is based on the Equilibrium Propagation method, which the authors apply to the DWAVE machine. I enjoyed reading the paper. It is very well written, the results novel and sound. The paper opens up a new application domain for Ising machines beyond combinatorial optimisation. Therefore, I believe the results are interesting to a broad community both for machine learning specialists as well as the large numbers of teams that are now building Ising machines. I also specifically liked the very careful assessment of the possibility of a quantum advantage.

Ising machines have been influential in the very beginnings of AI in the unsupervised training of generative networks, especially in the framework of Boltzmann machines. The authors touch upon this in their introduction. Although, it’s not clear if Boltzmann machines underperform due to the difficulty of correctly sampling the Boltzmann distribution and therefore having to rely on approximate methods for this. I think this is an important point to highlight better as the authors compare their results to unsupervised contrastive divergence learning on DWAVE.

We have revised the text to emphasize the two primary factors that contribute to the reduced accuracy of Boltzmann Machines compared to state-of-the-art (SOA) algorithms for classification tasks:

“However, Boltzmann machines are generative models that do not directly optimize a cost function related to a classification error. Furthermore, the parameters are typically evolved with approximations of the gradient, as the exact value is complicated and lengthy to compute. Boltzmann machines therefore underperform in difficult classification tasks when compared to standard supervised learning algorithms like backpropagation [cite{krizhevsky_convolutional_nodate}].”

The authors refer to Ising machines as binary spin systems. A lot of the properties that the authors refer to are also related to binary spin systems. However, the umbrella term of Ising machines encompasses much more. There are classical systems such as from the Rome group (Pierangeli2019): [ref: https://journals.aps.org/prl/pdf/10.1103/PhysRevLett.122.213902](https://journals.aps.org/prl/pdf/10.1103/PhysRevLett.122.213902). There is also a large variety of analogue Ising machines implementing the spins not as binary values but as analogue values e.g. Ising machines developed by Stanford (McMahon2016) <https://www.science.org/doi/full/10.1126/science.aah5178>, NTT (Inagaki,2016) <https://www.science.org/doi/full/10.1126/science.aah4243>, and many more. No mention is made of these works in the paper, while the authors do refer to the distinct advantages of the DWAVE.

We have modified the text in the introduction to underline that both digital and analog hardware can be used to realize Ising machines that implement the same Ising Hamiltonian of coupled spins:

“Ising Machines \cite{mohseni_ising_2022} are analog \cite{McMahon2016,Pierangeli2019,cai_power-efficient_2020,Litvinenko2023} or digital hardware systems \cite{tsukamoto2017accelerator,Lo2023} that are particularly fitted for this purpose, as they are designed to find the ground state of the Ising spin model”

It is therefore unclear to what they compare. As an example the authors write that most Ising machines have local connectivity. This is true for DWAVE, but the systems mentioned above can have full connectivity.

We have modified the text in the introduction in order to clarify our contribution in the context of all the existing hardware Ising Machines, with parallel or sequential dynamics, and local or full connectivity:

“Third, the implementation of EP on Ising machines necessitates navigating the balance between connectivity and parallelism. In contrast to biological neural networks, which are highly interconnected systems where neurons evolve simultaneously, Ising machines typically fall into two distinct categories. The first category offers full connectivity but operates with sequential dynamics, leveraging measurement-feedback mechanisms for simulating the spin dynamics \cite{McMahon2016,yamamoto_coherent_2017}. Conversely, the second category showcases fully parallel dynamics but is limited by its sparser physical connections \cite{Aadit2022}. While the former is preferred for combinatorial optimization in current applications, the natural parallel dynamics towards an equilibrium state in the latter is especially fitting for Equilibrium Propagation. Strategies then need to be established to adapt the network architecture to the Ising hardware's connectivity.”

In the same context, the Brussels group have shown (in Bohm2021) <https://www.nature.com/articles/s41467-022-33441-3> that Ising machines can be used for unsupervised learning on the MNIST and it would be interesting to see a comparison to this work.

In Reference Bohm2021, the authors train their Boltzmann machine (BM) on smaller images (8x8 pixels) but use a larger dataset (>7000 images) than we do. They employ a sigmoidal activation function for neurons and must sample their Ising machine 1000 times per input to approximate this continuous function. By contrast, we sample only 10 times during training and use binary spin-like activations. Most notably, the Bohm2021 approach initially trains the BM's hidden layer unsupervised and later adds a supervised linear classifier trained on a digital computer. We accomplish all supervised training on the Ising machine in a single step.

We have included this discussion in the text:

“In Reference \cite{Bhm2022}, the authors trained a Boltzmann machine on an optical Ising machine, achieving ~95% accuracy on down-sampled MNIST (8X8). Our approach differs on several points. Their setup uses an FPGA for vector-matrix multiplication and adds a linear classifier on top of a hidden layer, trained separately. We train the full network, including the classifier, directly on the Ising machine using Equilibrium Propagation. Moreover, they use smaller images and more data, sampling their machine 1000 times per problem. In contrast, we sample just 10 times during training and demonstrate that a single sample suffices for post-training inference with binary neurons.”

In the description of the algorithm, there is a reliance on the tendency of Ising machines to reach the ground state. However, for many graphs, Ising machines tend to get stuck in local minima instead of the ground state. How big of an issue would this be?

The original algorithm of EP does not seek to find the ground state of the energy function, but just a steady equilibrium state of the system. Therefore the tendency of Ising machines to get stuck in local minima for certain graphs does not constitute an issue.

We have modified the section in the introduction related to EP to clarify this point:

“The algorithm requires that the physical system evolves towards a stable equilibrium state - that does not need to be the ground state \cite{scellier_equilibrium_2017} - through the minimization of an energy function”

Finally, the discussion of the Ising machine reaching a more deterministic regime (in Fig 4): this is something I’m quite unsure of how one would interpret this. Isn’t this more likely related to the task that is being trained. I.e. a task with a more simple energy landscape where the DWAVE always finds the ground state and by this removing the stochastics. Maybe, the authors can expand on this discussion. e.g. would this carry over to other tasks?

We clarify below the text of the article to highlight that the deterministic regime reached during training by EP is indeed an intrinsic feature of the algorithm, which sculpts the energy landscape to reach a pronounced energy minimum for every input, thus reducing the effective stochasticity for the inputs in the database corresponding to the different classes to recognize:

“Lastly, Ising machines are inherently stochastic in nature \cite{hamerly_experimental_2019}. D-Wave for instance operates at finite non-zero temperature, resulting in thermal excitation competing with quantum annealing. To solve combinatorial problems using this machine, the state needs to be sampled multiple times to obtain an accurate solution. Similarly, during our training process, we sampled the machine's state 10 times at the end of each phase to ensure optimal accuracy. Interestingly, post-training with Equilibrium Propagation, the D-Wave Ising machine's inherent randomness transitions to a far more predictable behavior, as illustrated in Fig. \ref{fig:fig4}. With the aid of this algorithm, the training process effectively molds the once arbitrary energy landscape, establishing a pronounced energy minimum near the ground state for each database input. We find that after training, the solution is given in a single call to the Ising machine, greatly reducing the inference time. Given that this attribute stems from the core mechanics of EP, we anticipate its applicability across diverse classification tasks. These results also open up the possibility of learning combinatorial problems through a data-driven approach that can provide faster and more accurate solutions than the traditional approach where Ising system parameters are defined by the problem.”

Minor comments:

- the spreads in Fig 2c are not very visible
- the text contains some typos and some references are incomplete
- the labels of fig 3a labels are very small

We thank the reviewer for pointing out all these typos, we updated the manuscript accordingly.

Reviewer #3 (Remarks to the Author):

In "Training an Ising Machine with Equilibrium Propagation", the authors use a commercially available D-Wave quantum annealer to perform a rudimentary learning task through the use of a much smaller version of the celebrated MNIST dataset. The authors use the Equilibrium Propagation (EP) algorithm.

Despite enjoying the originality of the manuscript as an impressive hardware experiment (where a good portion of the algorithm is performed on a digital computer), I have fundamental concerns about the eventual significance of such a study. Below are my general concerns:

[1] EP is a physics-inspired learning algorithm that has now been around for 7 years and despite the authors' enthusiasm (based on references to mostly their own papers), it is unclear whether EP really delivers in reaching state-of-the-art performance in modern ML. The literature review the authors perform largely focuses on critiquing Boltzmann Machines, another algorithm, whose relevance is questionable, as the authors are well-aware. The authors might want to show independent evidence as to whether EP can match the performance of the actual state of the art in ML, which is a moving and tough target to match.

We added in the Supplementary materials a table with the most recent results obtained with Equilibrium Propagation from our group but also from external groups on various tasks, showing that Equilibrium Propagation is the energy-based model leading to the highest accuracy on current benchmark tasks including CIFAR100 (<https://openreview.net/forum?id=VLszAxAFGs>).

[2] Second issue is the extremely scarce experimental data in the present version. One excuse given is "limited access time" to the D-Wave machine, whose meaning is vague in a scientific context.

Moreover, the "fully connected" network results in Figure 2 contain only a few data points on an extremely toy problem. Crucial details about the minor graph embedding (MGE) procedure, the well-known "chain-breaking" between ferromagnetic spins are absent.

Chain-breaking often changes the "actual" ground state of an embedded problem which could affect the sensitive nature of relaxation in the EP algorithm. It is unclear if the authors are aware of this problem or paid attention to it because it is unaddressed.

We have used the simple majority vote procedure to address the chain breaking issue in our present work. Chain breaking is indeed a major problem when the ground state of the problem is required with high precision such as for doing combinatorial optimization. Interestingly, in its original formulation and implementation, EP does not require reaching the exact ground state of the system, a point that we have now clarified in the introduction:

"The algorithm requires that the physical system evolves towards a stable equilibrium state - that does not need to be the ground state \cite{scellier_equilibrium_2017} - through the minimization of an energy function"

Consequently, we anticipate that training neural networks with Equilibrium Propagation (EP) on Ising Machines will prove significantly more resilient to chain-breaking issues than when solving combinatorial problems. We have added a paragraph in the Methods section addressing this point.

“The embedding process relies heavily on the chaining procedure - using multiple hardware spins to represent a single spin in order to couple it to more neighbors than the architecture allows for - and the chaining strength can be adjusted. In practice, we set the chaining strength (i.e., the value of the couplings between the hardware spins that represent the same spin) to -1 which has proven effective. Moreover, spins within a single chain may not all align post-annealing, leading to 'chain breaking.' We address this through a straightforward majority-vote strategy. While chain breaking poses significant challenges in contexts requiring highly precise ground states, such as combinatorial optimization, Equilibrium Propagation (EP) merely necessitates reaching a stable equilibrium state \cite{scellier_equilibrium_2017}. As a result, we expect neural network training via EP on Ising Machines to be far more tolerant of chain-breaking complications compared to combinatorial problem-solving.”

Additionally, I have found the critical discussion regarding how the qubits are "stuck" in the nudging phase of the algorithm lacking. It sounds like there was a very delicate fine-tuning to get those few data points which would be exacerbated in real problems.

We have clarified this point in the Methods section:

“Following the initial implementation of EP \cite{scellier_equilibrium_2017}, we performed the two sequential phases of EP on the Ising machine. The input image remained static during all the annealing process, as required for the system to converge given a fixed input. We had to adjust the duration of the annealing for the free phase. We chose the native duration which is $20\ \mu\text{s}$ (see Fig.\ref{fig:fig1}b). For the reverse annealing we specified the duration and the schedule. We set the duration to $40\ \mu\text{s}$, the initial annealed fraction to 0, the annealed fraction at time $20\ \mu\text{s}$ to 0.25 and the final annealed fraction to 0 (see Fig.\ref{fig:fig1}b). This enabled the system to change its state according to the nudging signal. This optimal value for the annealed fraction, as determined midway through the reverse annealing process, was first calibrated using a simplified task and then applied to training on the MNIST/100 dataset. Specifically, we employed the same MLP (Multilayer Perceptron) architecture (784-120-40) but trained it on the MNIST/10 dataset. We incrementally adjusted the annealed fraction until the gradients registered on the chip exceeded zero, indicating effective training. This was further validated as the loss diminished and the accuracy exhibited an upward trend.”

[3] Third is the intentional or unintentional conflating of "Ising Machines" with the "D-Wave Machine". Most importantly, the D-Wave machine does NOT implement the classical Ising model but rather the transverse field Ising model (unfortunately not even mentioned even superficially). This difference may sound unimportant, however, the dynamics of the TFIM and its equivalence to the Ising model have been a topic of intense discussion.

We thank the referee for pointing out this issue. Indeed the Hamiltonian during annealing is the transverse field Ising model. In our study, when we measure the spin states on the Z axis to extract the EP weight updates, the system is in a classical state.

We have reformulated footnote 1 to avoid confusion:

“We use here an Ising energy that differs from the standard one by a minus sign, in order to match the D-Wave formulation \cite{harris_experimental_2010}, and to be consistent with all the equations and learning rules in the rest of the paper.”

(the previous version was: “We write here the Energy function that the D-Wave Ising machine really minimizes \cite{harris_experimental_2010}, which differs from the classical Ising Energy

function by a minus sign, in order to be consistent with all the equations and learning rules in the rest of the paper.”)

In addition, we have added the following note in Methods:

“We want to emphasize that while the quantum annealing procedure is implemented through a dynamical transverse Ising model [63] that allows for quantum fluctuations, the final state obtained through the same quantum annealing procedure is solution of the classical Ising model.”

The remarks about why the D-Wave Machine is the best available Ising Machine are simply not correct: despite its possible advantages that are only recently becoming clear [see <https://www.nature.com/articles/s41586-023-05867-2>], D-Wave is among the worst performing Ising Machines in optimization (see, Matthew Kowalsky et al 2022 Quantum Sci. T) due to the aforementioned chain-breaking, short coherence times and low bit precision. Same issues persist in sampling problems as well.

There are many other, larger Ising Machines that could be used for an EP-training on "Ising machines".

That the authors laud the bit precision of D-Wave while casting their study in the context of Ising Machines is quite surprising to put it mildly: I strongly recommend they change their title or actually train their algorithm using an IM, wherever this paper is published.

Please note that this is not a criticism of the D-Wave machine (which has already shown better scaling in select problems) but rather the strategic focus of the present manuscript on something it does not address.

We are using here the generic name “Ising Machine” to describe the D-Wave hardware, following a terminology often used in the field of hardware developments for optimization and neuromorphic computing , as in <https://www.nature.com/articles/s42254-022-00440-8> and articles cited therein.

We have modified the text of the introduction to highlight that different types of Ising machine hardware exist, analog or digital,

“Ising Machines \cite{mohseni_ising_2022} are analog \cite{McMahon2016,Pierangeli2019,cai_power-efficient_2020,Litvinenko2023} or digital hardware systems \cite{tsukamoto2017accelerator, Lo2023} that are particularly fitted for this purpose, as they are designed to approximate the ground state of the Ising spin model”

with parallel or sequential dynamics, and local or full connectivity. We clarify the interest of natural parallel dynamics, as supported by D-Wave, for implementing Equilibrium Propagation:

“Third, the implementation of EP on Ising machines necessitates navigating the balance between connectivity and parallelism. In contrast to biological neural networks, which are highly interconnected systems where neurons evolve simultaneously, Ising machines typically fall into two distinct categories. The first category offers full connectivity but operates with sequential dynamics, leveraging measurement-feedback mechanisms for simulating the spin dynamics \cite{McMahon2016,yamamoto_coherent_2017}. Conversely, the second category showcases fully parallel dynamics but is limited by its sparser physical connections \cite{Aadit2022}. While the former is preferred for combinatorial optimization in current applications, the natural parallel dynamics towards an equilibrium state in the latter is

especially fitting for Equilibrium Propagation. Strategies then need to be established to adapt the network architecture to the Ising hardware's connectivity.”

[4] The final problem I feel unaddressed is the following:

- why should we go through the difficult procedure of minor-graph embedding, securing a D-Wave annealer, and having a hybrid classical computer to perform the gradient descent and input multiplication, as done by the authors?
- Why would a software simulation of simulated annealing (as the authors have done in Figure 2) not be sufficient for whatever purpose we have? The authors seem to be talking about "low power" and "intrinsic dynamics" but the question is really not answered.

Our work demonstrates that we can train an actual unconventional hardware with Equilibrium Propagation, even if it was not initially designed to, by exploiting its ability to minimize its own energy.

- The graph embedding process is performed only once at the start of the training, and this embedding is consistently used throughout the training duration. This means the potentially resource-intensive graph embedding step gets spread out, making it more cost-effective over numerous training iterations. As highlighted in the section discussing the convolutional neural network, we anticipate that future architectures driven by layout (which are entirely independent of the graph embedding procedure) will offer greater scalability.

We have added this paragraph in the Methods section “5.3.1 Embedding the fully-connected architecture onto the chip layout” to clarify the embedding procedure and its impact in energy consumption.

- We have introduced methods that are adaptable to any annealing-based Ising machines. Our present methodology relies on a digital computer for parameter storage and updates. However, advancements are underway in dedicated hardware implementations, which can manage these updates in-situ, rendering external computers redundant [<https://www.nature.com/articles/s41928-020-0436-6>]. These emerging hardware solutions promise energy consumption reductions of up to 100 times compared to GPUs. This makes them a more energy-efficient alternative to software simulations of such systems, like those using simulated annealing [<https://www.nature.com/articles/s41586-023-06337-5>].

We have modified the discussion to highlight this point:

“In line with this trend, a recent study \cite{Yi2022} successfully trained a crossbar array of memristors to emulate the couplings of an Ising-like model using a learning law similar to Equilibrium Propagation. Contrary to our study, the system in this research is not intrinsically dynamic. Instead, the "spin" dynamics is emulated digitally in discrete time by iteratively and recursively interfacing the memristive system to digital electronics. Combining our approach, in which an Ising spin system computes gradients through its intrinsic dynamical evolution towards a minimum of energy, with the use of memristive couplings presents significant potential for future low-power, embedded Ising machines that can be trained on the edge.”

Our findings demonstrate that Ising machines are not limited to combinatorial optimization; they can also be trained for AI tasks. This paves the way for the creation of ultra-energy-efficient Ising Machines utilizing emerging technologies for the implementation of spin couplings. By harnessing the system's intrinsic dynamics for weight computation and leveraging memory-in-computing for weight storage, we can achieve substantial reductions in energy consumption.

The authors' claim about how their D-Wave implementation "beats simulated annealing" in every task without a clear discussion of hyper-parameters is very flimsy. I strongly doubt their D-Wave training method can withstand a properly optimized SA, and this is hard to tell with the scant details in the paper.

We have modified the text to clarify that indeed, we can only conclude that the hardware obtains software-equivalent results:

"Our experiments were carried out using the D-Wave machine, whose quantum properties are a topic of ongoing discussion in the scientific community [48, 49]. Our work, however, is fundamentally classical and could be applied to any Ising Machine with the capacity to stabilize in its energy minimum. Although we found that the accuracy obtained on the hardware through quantum annealing is slightly better than software simulations, we cannot conclude on a quantum advantage as the exact training conditions of the hardware cannot be easily replicated in simulations (see Methods). To establish more definitive conclusions on this matter, it would be valuable to investigate whether the D-Wave machine maintains higher accuracy compared to software simulations when trained on other tasks (especially those necessitating an optimal ground state for the Ising system), and ideally, to compare the hardware accuracy with that of classical Ising machines with the same connectivity."

REVIEWER COMMENTS

Reviewer #1 (Remarks to the Author):

Thanks for the responses, which clarified the issues I raised.

Reviewer #2 (Remarks to the Author):

The authors have answered my questions and remarks in a convincing way. They have improved their manuscript. I'm still very much in favour of acceptance.

Reviewer #3 (Remarks to the Author):

In my original review, I raised a few significant issues, namely:

{1} The significance and relevance of the Equilibrium Propagation (EP) algorithm

As a response, the authors pointed to a workshop paper, stating "Equilibrium Propagation is the energy-based model leading to the highest accuracy". The first point is to note that the paper the authors cited takes a more balanced approach in comparing energy-based learning algorithms. The second point is the data presented in the supplementary, say compared to Back Propagation Through Time, does not seem to be that different from Equilibrium Propagation. As such, the authors seem to be somewhat overstating their case, though I lift my disbelief regarding the EP algorithm, given this prior work. Even then, it would be good to provide genuine comparisons.

{2} Problems of minor graph embedding (chain-breaking, costs of MGE)

As a response, the authors responded by saying that EP does not need to reach absolute ground states, which begs the question of why one would need a D-Wave annealer in the first place. The points about

MGE being "amortized" over training is not convincing because the authors solution points to "future implementations driven by layout", once again, begging the question of why one would need D-Wave in the first place. Moreover, the authors now clarify their position regarding "quantumness" of the D-Wave machine saying that they use it as a classical machine. In that case, it is not clear what role quantum annealers play in this work. Why not use widely available classical Ising machines, if all the issues with QAs seem to be problems to be solved later?

{3} My final question was the following: Why would a software simulation of simulated annealing (as the authors have done in

Figure 2) not be sufficient for whatever purpose we have? The authors seem to be talking about "low power" and "intrinsic dynamics" but the question is really not answered.

The answer seems to repeat the importance of "intrinsic dynamics" (rather than prior FPGA or CMOS implementations that are classical and now curiously called "simulations"). FPGAs aside, since they are rather general-purpose, it is not clear to me why using the intrinsic physics of digital CMOS is fundamentally different compared to using D-Wave. "Analog" is appealing only if it serves some purpose. Here, the authors mention memristors or other technology for low-power implementations. Unfortunately, everything is in the future.

Finally, I raised concerns about why software simulations on classical simulated annealing should not drastically outperform the results presented. The authors responded by saying "they cannot conclude a quantum advantage". I think this is changing the subject. The authors should focus on concluding whether there is any classical advantage or not. Since they themselves are now in agreement that there is not much that is quantum about any of their results ...

I do not want to be too negative, this is a nice piece of work that deserves to be published somewhere, with tempered and straightforward claims. But I do not see why this paper needs to be in Nature Communications with these serious issues that were not really addressed.

Response to reviewers

We thank all the reviewers for their appreciation of our work and thought-provoking questions. Our responses are in blue, and the new changes to the text in orange.

Reviewer #1 (Remarks to the Author):

Thanks for the responses, which clarified the issues I raised.

Reviewer #2 (Remarks to the Author):

The authors have answered my questions and remarks in a convincing way. They have improved their manuscript. I'm still very much in favor of acceptance.

Reviewer #3 (Remarks to the Author):

In my original review, I raised a few significant issues, namely:

{1} The significance and relevance of the Equilibrium Propagation (EP) algorithm

As a response, the authors pointed to a workshop paper, stating "Equilibrium Propagation is the energy-based model leading to the highest accuracy". The first point is to note that the paper the authors cited takes a more balanced approach in comparing energy-based learning algorithms. The second point is the data presented in the supplementary, say compared to Back propagation Through Time, does not seem to be that different from Equilibrium Propagation. As such, the authors seem to be somewhat overstating their case, though I lift my disbelief regarding the EP algorithm, given this prior work. Even then, it would be good to provide genuine comparisons.

We thank the reviewer for highlighting the need to deepen the comparison between EP and other algorithms.

- We clarify in the introduction why BPTT is the benchmark for EP for training convergent recurrent energy-based models: EP approximates the exact gradients provided by BPTT.

"Introduced in 2017, Equilibrium Propagation (EP) \cite{scellier_equilibrium_2017} has garnered significant attention for its ability to train in a supervised way convergent recurrent energy-based models. Unlike traditional methods that compute gradients of the objective function using Backpropagation Through Time (BPTT), EP employs a local learning rule that not only approximates BPTT-derived gradients \cite{NEURIPS2019_67974233} but also overcomes the limitations of conventional training in physical systems \cite{kendall_training_2020,martin_eqspike_2020, dillavou-PRA, Yi2022}."

- We provide in the Supplementary materials a table that compares the reported accuracies for different hardware-compatible algorithms* on the benchmark image classification task CIFAR-10. The table shows that the test error drops with the number

of trained layers, the best results being obtained with 16 layers. For the same network architecture of 4 convolutional layers, EP achieves only 1% less than the exact gradients provided by BPTT.

	PEPIT A [Delaferra2022]	FA [Nokland2019]	DFA [Nokland2019]	Hebbian learning without feedback [Journé2023]	Coupled Learning [Scellier2023]	EP [Scellier23]	Recurrent BP [Scellier23]	BPTT [Scellier2023]	TargetProp [Ernout2023]	Sigprop [Kohann2022]	CHL diadic neurons - Dual Prop [Hoeir2022]
Architecture	1 conv layer	3 conv layers	3 conv layers	3 conv layers	4 conv layers	4 conv layers	4 conv layers	4 conv layers	5 conv layers	VGG 8	VGG 16
Test error (%)	43.67	27.1	26.9	19.7	13.5	11.1	10.7	10.1	10.4	8.34	7.7

* ie with local learning rules (based on information available to the parameter-to-be-updated instead of rules based on chain-rule derivation) and/or dynamical neurons (closer to the actual hardware).

2} Problems of minor graph embedding (chain-breaking, costs of MGE)

As a response, the authors responded by saying that EP does not need to reach absolute ground states, which begs the question of why one would need a D-Wave annealer in the first place. The points about MGE being "amortized" over training is not convincing because the authors solution points to "future implementations driven by layout", once again, begging the question of why one would need D-Wave in the first place. Moreover, the authors now clarify their position regarding "quantumness" of the D-Wave machine saying that they use it as a classical machine. In that case, it is not clear what role quantum annealers play in this work. Why not use widely available classical Ising machines, if all the issues with QAs seem to be problems to be solved later?

We have added in Supplementary materials the table that we also include below, indicating different types of Ising machines that have been reported, and their specificities. As can be seen from the table, most Ising Machines are lab-based experimental set-ups that are not publicly available. The only two publicly available Ising machines, controllable via command lines through an interface, are the Fujitsu / Toshiba FPGA-based IM and the D'Wave superconducting-qubits-based IM.

Our choice of D'Wave with respect to the FPGA-based IM is not driven by a comparison between quantum and classical computing, but by its advantages for training with EP.

The first advantage highlighted by the table is technical: contrary to the FPGA-based IM, D'Wave has a good application programming interface, which is crucial for training an IM.

The second advantage is fundamental and encompasses the meaningfulness of our work: contrary to the FPGA-based IM, with D'Wave, there is a match between the physics of the hardware system and the physics of the training algorithm.

FPGA-based IM implement numerical simulations - ie Glauber dynamics - of the spin dynamics that are compiled on digital CMOS. There are no physical hardware spins in this IM. The CMOS transistors in the FPGA do not minimize the Ising energy.

The D'Wave IM is based on unconventional hardware nanodevices. The spin qubits are physical SQUIDS that are physically coupled through Josephson junctions. The coupled spin qubits do minimize the Ising energy, intrinsically.

Our work belongs to the growing field of Physical neural networks [Wright et al., Nature 2023], where the goal is to develop physical systems based on unconventional nanodevices that solve AI tasks through the natural laws of physics [Markovic et al., Nature Reviews Physics 2020][Torrejon et al., Nature 2017][Kumar et al., Nature Review Physics, 2022][Kiralý et al., Nature nanotechnology, 2021]. Our results show that a physical system of coupled spins can learn to perform supervised AI tasks through an algorithm that harnesses its intrinsic ability to minimize an energy, arising from the natural laws that govern our physical world.

We have modified the introduction to clarify our contribution:

“Our work belongs to the growing field of Physical neural networks [Wright et al., Nature 2023], where the goal is to develop physical systems based on unconventional nanodevices that solve AI tasks through the natural laws of physics [Markovic et al., Nature Reviews Physics 2020][Torrejon et al., Nature 2017][Kumar et al., Nature Review Physics, 2022][Kiralý et al., Nature nanotechnology, 2021]. We aim to show that a physical system of coupled spins can learn to perform supervised AI tasks through an algorithm that harnesses its intrinsic ability to minimize an energy, arising from the natural laws that govern our physical world.”

Table 1: Different hardware Ising machine implementations and main relative features

Technology	Physical encoding of the Ising energy	Implementation of the spin dynamics	Connectivity (implementation)	Previous use	ML algorithm and task solved with the IM	Availability/ interface
Coherent Ising Machine [Wang et al., PRA, 2013] [Yamamoto et al. NPJ, 2017]	Yes: Optical loss	- Optical pulses encoding spins in the phase - non-linear dynamics with phase-dependent gain - time-multiplexed	All-to-all (up to 100 000 spins [Honjo et al. Science Advances, 2021]) (Measurement-feedback couplings with a side FPGA)	Combinatorial optimization	Non reported	Limited: experimental work - unknown
Poor's man Ising machine [Bohm et al., Nature Comm., 2019]	No	- Mixed digital-optical implementation: discrete Euler updates realized through a non-linear electro-optical modulator. - time-multiplexed	All-to-all (1 000 spins) (Measurement-feedback couplings with a side FPGA)	- MCMC acceleration - ML: Pre-trained a Boltzmann machine on the hardware but require backprop to	Contrastive divergence for training a Restricted Boltzmann machine (sub-sampled MNIST) [Bohm et al., Nature Comm., 2019]	Limited: experimental work - unknown

				fine-tune the model		
Memristive IM (Strachan) [Cai et al., Nature Electronics, 2020]	No	- Digital spins updated with hysteretic thresholding the current flowing from the crossbar array - parallel updates	All-to-all (60 spins) (Direct coupling through a crossbar array)	Combinatorial optimization	MADeM (close to EqProp) for training the network as a Hopfield network (Reconstructing and classifying braile sentences) [Yi et al., Nature Electronics 2023]	Limited: experimental work - unknown
FPGA (Fujitsu) [Tsukamoto et al. Technical report, 2017]	No	- Digital spins updated according to Glauber dynamics - individual updates	All-to-all (8 192 spins) (Measurement-feedback couplings)	Combinatorial optimization ML: unknown use	Non reported	Yes: commercially available as a remote ressource (cloud) - poor API - no python interfacability
FPGA (Toshiba) [Tatsumura et al. ICFPLA 2019]	Simulated loss of coupled oscillators (simulated Coherent Ising Machine)	- Digital spins with simulated bifurcation dynamics	All-to-all (8-coupled FPGAs: 8 192 spins [Tatsumura, HEART, 2021], GPU: under development) (Measurement-feedback couplings)	Combinatorial optimization ML: unknown use	Non reported	Yes: commercially available as a remote ressource (cloud) - poor API - no python interfacability
CMOS [Lo et al., Nature Electronics, 2023]	Yes: Phase interaction (Kuramoto model)	- Spins encoded in the phase of the electrical oscillators - parallel dynamics	All-to-all (48 spins) (Direct coupling)	Combinatorial optimization	Non reported	Limited: experimental work - unknown
Superconducting HW (DWAVE) [Harris et al., PRB, 2010]	Yes: Hamiltonian of the system	- Spins encoded in the circulation of a superconducting current - parallel dynamics	Local: Chimera (2042 spins) or Pegasus (5640 spins) (Couplings through local SQUIDs)	Combinatorial optimization ML: - Adachi et al. pre-trained a Boltzmann machine on sub-sampled MNIST on it but still requires fine-tuning with backprop afterwards	Contrastive divergence for training a Restricted Boltzmann machine [Adachi et al., arxiv, 2015]	Yes: commercially available as a remote ressource (cloud) - Python API with calls to the solver - very simple interface

{3} My final question was the following: Why would a software simulation of simulated annealing (as the authors have done in Figure 2) not be sufficient for whatever purpose we have? The authors seem to be talking about "low power" and "intrinsic dynamics" but the question is really not answered. The answer seems to repeat the importance of "intrinsic dynamics" (rather than prior FPGA or CMOS implementations that are classical and now curiously called "simulations"). FPGAs aside, since they are rather general-purpose, it is not clear to me why using the intrinsic physics of digital CMOS is fundamentally different compared to using D-Wave. "Analog" is appealing only if it serves some purpose. Here, the authors mention memristors or other technology for low-power implementations. Unfortunately, everything is in the future.

We would like to emphasize that while the progress of standard digital computing has constantly sustained the progress of AI, the end of Moore's law [Chien et al., IEEE Computer, 2013] questions the future scaling of digital computing. Moreover, using standard digital processors for running AI models (for both training and inference) does not come for free: such machines are built on the Von Neumann architecture [Von Neumann, ENIAC Scientific Report, 1945] where the memory and the processing unit are physically separated. The millions / billions of parameters constantly need to be moved back and forth from memory to the processing unit to realize the operations required in neural networks. This Von Neumann "bottleneck" is both very costly in terms of energy [Chen et al., ISCA 2016]

Operation	Energy consumption
Addition of data	1x
Access data (onchip cache)	60x
Access data (offchip RAM)	3500x

from [Pedram et al , IEEE D&T 2016]

and slows down dramatically the algorithm as data transfer is ultimately limited by the maximal frequency of the hardware (currently stagnating to a few GHz for current processors) [Markov, Nature , 2014][Zhang et al., Nature Electronics, 2020].

These features of standard digital computing motivate the search for alternative hardware to compute. It is today recognized that they will rely on the use of unconventional nanodevices (NVM in the graph below) that circumvent the Von Neumann "bottleneck" by storing data where the computation takes place (in-memory computing)[Ambrogio et al., Nature 2018][Zidan et al., Nature Electronics, 2018][Sebastian et al., Nature Nanotechnology, 2020][Harabi et al. Nature Electronics 2022].

From [Zhang et al., Nature Electronics 2020]

The development of unconventional hardware naturally questions the models and the corresponding algorithms to be run on it [Jaeger et al., Nature Comms 2023]. For the specific case of hardware or physical neural networks, backpropagation is difficult to realize end-to-end in hardware without major overhead costs (massive peripheral circuitry and memory) [Markovic et al., Nature Reviews Physics 2020][Torrejon et al., Nature 2017][Kumar et al., Nature Review Physics, 2022][Kiraly et al., Nature nanotechnology, 2021].

New learning algorithms grounded in the physics of the hardware are emerging. EP is one of them but we could cite Hamiltonian Echo Backprop [Lopez-Pastor, PRX 2023], Coupled Learning [Stern et al, PRX 2021], Thermodynamics computing [Coles et al. arXiv:2302.06584][Aifer et al., arxiv:2308.05660, 2023], Forward-forward-like algorithms [Momeni et al., Science 2023], Deep reservoir computing [Gallicchio et al., Neurocomputing, 2017][Gauthier et al., Nature Comms., 2021].

Hardware demonstrations of those alternative training algorithms are milestones sought after by the unconventional computing community as it embodies the algorithm in actual hardware rather than abstract ideal software simulations.

Our work is one of those hardware implementations of an alternative to backpropagation learning algorithm (EP) on actual hardware. We show that matching the hardware (a physical system of coupled spins evolving according to the Ising energy - D'Wave system) with the algorithm (a training algorithm harnessing the energy minimization of an Ising energy to find weight updates) is an efficient way to achieve learning in unconventional hardware. Future work could use the methods we have developed here on other lower power (maybe faster) hardware.

We have modified the discussion to clarify our contribution to the field of training unconventional hardware:

“Our results and the algorithm employed to obtain them can be applied to any type of annealing-based Ising machine. Those with an ultra-low power consumption are particularly appealing for reducing the overall electrical consumption of AI and deploying it in embedded systems. Memristors or spintronic nano-components are currently being extensively researched as building blocks for such systems, as they enable the co-integration of memory, novel physical functionality and computing. This greatly enhances the efficiency and scalability of the system, making it more suitable for real-world applications.

The development of unconventional hardware naturally questions the models and the corresponding algorithms to be run on it [Jaeger et al., Nature Comms 2023]. For the specific case of hardware or physical neural networks, backpropagation is difficult to realize end-to-end in hardware without major overhead costs (massive peripheral circuitry and memory) [Markovic et al., Nature Reviews Physics 2020][Torrejon et al., Nature 2017][Kumar et al., Nature Review Materials, 2022][Kiraly et al., Nature nanotechnology, 2021].

New learning algorithms grounded in the physics of the hardware are emerging such as Hamiltonian Echo Backprop [Lopez-Pastor, PRX 2023], Coupled Learning [Stern et al, PRX 2021], Thermodynamics computing [Coles et al. arXiv:2302.06584][Aifer et al., arxiv:2308.05660, 2023], Forward-forward-like algorithms [Momeni et al., arXiv:2304.11042, 2023], Deep reservoir computing [Gallicchio et al., Neurocomputing, 2017][Gauthier et al., Nature Comms., 2021] and EP. Hardware demonstrations of those alternative training algorithms are milestones sought after by the unconventional computing community as it

embodies the algorithm in actual hardware rather than abstract ideal software simulations [Wright et al., Nature 2023] [Markovic et al., Nature Reviews Physics 2020][Torrejon et al., Nature 2017][Kumar et al., Nature Review Physics, 2022][Kiraly et al., Nature nanotechnology, 2021].

In line with this trend, a recent study \cite{Yi2022} successfully trained a crossbar array of memristors to emulate the couplings of an Ising-like model using a learning law similar to Equilibrium Propagation. Contrary to our study, the system in this research is not intrinsically dynamic. Instead, the "spin" dynamics is emulated digitally in discrete time by iteratively and recursively interfacing the memristive system to digital electronics.

We show that matching the hardware (a physical system of coupled spins evolving according to the Ising energy - D'Wave system) with the algorithm (a training algorithm harnessing the energy minimization of an Ising energy to find weight updates) is an efficient way to achieve learning in unconventional hardware. Future work could use the methods we have developed here on low power and faster embedded hardware."

Finally, I raised concerns about why software simulations on classical simulated annealing should not drastically outperform the results presented. The authors responded by saying "they cannot conclude a quantum advantage". I think this is changing the subject. The authors should focus on concluding whether there is any classical advantage or not. Since they themselves are now in agreement that there is not much that is quantum about any of their results ...

Training neural networks on hardware Ising machines has recently become a very active research field [Adachi et al., arXiv:1510.06356, 2015][Khoshaman et al., Quantum Science and Technology, 2018][Bohm et al., Nature Comms., 2019][Job et al., arXiv:2009.00134][Niazi et al., arXiv:2303.10728, 2023], due to the promise of higher speed and energy efficiency. For such an approach to become mainstream however, it is important to find methods that maintain an accuracy competitive with current methods.

The two sources of inaccuracies when training an IM are:

- The algorithm. Supervised algorithms with a global loss function are state-of-the-art. But IMs are trained with unsupervised algorithms that do not reach the accuracy of BP or BPTT, even in digital simulations, because they do not optimize a global loss function. Furthermore, RBMs trained on IMs represent only a pre-training step, with all papers subsequently fine-tuning the model using backpropagation.
- The hardware itself. Hardware IMs have limited weight precisions, and those that are not based on digital CMOS are prone to noise. It is important to find training algorithms that maintain a high accuracy despite those imperfections.

Our study tackles these two issues.

We show that that we can train Ising machines with a supervised algorithm that optimizes a global loss function: EP

We show that the trained hardware IM reaches an accuracy on par with digital simulations (gradient based dynamics and simulated annealing) despite hardware non-idealities.

We attribute this result to the ideal match between the energy-based nature of physical Ising systems and the energy-based nature of the EP algorithm that encodes a global loss function physically.

We do not study nor conclude on classical versus quantum advantages. The methods that we develop can be applied to any annealing-based hardware IM.

We have modified the discussion to clarify this point:

“Although we found that the accuracy obtained on the hardware through quantum annealing is slightly better than software simulations, we cannot conclude on a quantum or classical advantage as the exact training conditions of the hardware cannot be easily replicated in simulations (see Methods).”

I do not want to be too negative, this is a nice piece of work that deserves to be published somewhere, with tempered and straightforward claims. But I do not see why this paper needs to be in Nature Communications with these serious issues that were not really addressed.

REVIEWERS' COMMENTS

Reviewer #3 (Remarks to the Author):

The authors have made substantial changes to their manuscript. Given Julie Grollier's track record and the meaningful and thoughtful responses they have given, I am almost convinced and the review process is converging.

Two comments:

(1) I am still confused about the response given to the simple question of we cannot have a Simulated Annealing comparison (to which, the answer is a generic "Moore's Law is Dead, so a simple experiment with D-Wave's QA is the solution" type of answer.

The reason I am confused is the authors DO have a simulated annealing comparison D-Wave with 5.5 -- I would make sure these are clearly pointed to in the main manuscript.

(2) My one remaining comment is about this response:

"The second advantage is fundamental and encompasses the meaningfulness of our work: contrary to the FPGA-based IM, with D'Wave, there is a match between the physics of the hardware system and the physics of the training algorithm. FPGA-based IM implement numerical simulations - ie Glauber dynamics - of the spin dynamics that are compiled on digital CMOS. There are no physical hardware spins in this IM. The CMOS transistors in the FPGA do not minimize the Ising energy."

This is an unfortunate statement, perpetuating a known misconception of the field, often by device engineers. There ARE physical spins in a CMOS or FPGA-based Ising Machine, just like there are physical spins in D-Wave. In CMOS, these spins have PRNGs and look up tables and other types of modular building blocks. One can point to these silicon transistors, draw a black box around them, and call them spins. The only difference is that they are mostly digital (though they do have analog noise).

Here is a thought experiment for the authors to consider: if we generate a stochastic bitstream of 1's and 0's from a CMOS-implementation of a spin, and a thresholded D-Wave implementation of a spin, could they tell the difference between which bitstream comes from which source?

If the answer is no, then one should not mystify or artificially provide some meaning to the more "analog" looking hardware. This type of argument is often used by proponents of analog hardware in order to rule out perfectly equivalent digital counterparts.

If the answer is yes, the authors should explain what it is that they mean by "physical systems that solve problems through the natural laws of physics" since, they are well-aware, CMOS transistors and FPGAs also operate based on the natural laws of physics.

Please note that I am not referring to highly synthesized and compiled hardware. What I am referring to is systems that are modularly developed from micro-architecture, where physical spins are identifiable and regular, in CMOS hardware.

The authors should make clarifying statements to their paragraph where Wright's paper is cited so that they do not mislead the field into NOT providing comparisons to the types of IMs developed by many groups and companies in the world (Fujitsu, Toshiba, ... etc.).

The rest looks good, and after these minor points are taken into account, given the other reviewer suggestions, I am happy to recommend publication.

Response to reviewers

Reviewer #3 (Remarks to the Author):

The authors have made substantial changes to their manuscript. Given Julie Grollier's track record and the meaningful and thoughtful responses they have given, I am almost convinced and the review process is converging.

We thank the reviewer for the appreciation of our work and thought-provoking comments and questions. Our responses are in blue, and the new changes to the text in red.

Two comments:

(1) I am still confused about the response given to the simple question of we cannot have a Simulated Annealing comparison (to which, the answer is a generic "Moore's Law is Dead, so a simple experiment with D-Wave's QA is the solution" type of answer.

The reason I am confused is the authors DO have a simulated annealing comparison D-Wave with 5.5 -- I would make sure these are clearly pointed to in the main manuscript.

We have clarified the comparison with Simulated Annealing in the main text of the manuscript:

"In Fig. 2c-d, we compare the accuracy reached by the physical system to numerical simulations. The first network (dashed lines in Fig. 2c-d) is a spin network identical to the one on the chip, and trained in the same way by replacing the quantum annealing with Simulated Annealing (SA-EP). The second network (solid lines in Fig. 2c) is a software Artificial Neural Network with binary activations and real-value weights (ANN-EP) evolving according to a Hopfield energy and trained by Equilibrium Propagation. The Simulated Annealing along with the Artificial Neural Networks (ANN) are executed on a digital processor as detailed in Methods. Consequently, they establish the benchmark for accuracy that we aim to achieve."

(2) My one remaining comment is about this response:

"The second advantage is fundamental and encompasses the meaningfulness of our work: contrary to the FPGA-based IM, with D-Wave, there is a match between the physics of the hardware system and the physics of the training algorithm. FPGA-based IM implement numerical simulations - ie Glauber dynamics - of the spin dynamics that are compiled on digital CMOS. There are no physical hardware spins in this IM. The CMOS transistors in the FPGA do not minimize the Ising energy."

This is an unfortunate statement, perpetuating a known misconception of the field, often by device engineers. There ARE physical spins in a CMOS or FPGA-based Ising Machine, just like there are physical spins in D-Wave. In CMOS, these spins have PRNGs and look up tables and other types of modular building blocks. One can point to these silicon transistors, draw a black box around them, and call them spins. The only difference is that they are mostly digital (thought they do have analog noise).

Here is a thought experiment for the authors to consider: if we generate a stochastic bitstream of 1's and 0's from a CMOS-implementation of a spin, and a thresholded D-Wave

implementation of a spin, could they tell the difference between which bitstream comes from which source?

If the answer is no, then one should not mystify or artificially provide some meaning to the more "analog" looking hardware. This type of argument is often used by proponents of analog hardware in order to rule out perfectly equivalent digital counterparts.

If the answer is yes, the authors should explain what it is that they mean by "physical systems that solve problems through the natural laws of physics" since, they are well-aware, CMOS transistors and FPGAs also operate based on the natural laws of physics.

Please note that I am not referring to highly synthesized and compiled hardware. What I am referring to is systems that are modularly developed from micro-architecture, where physical spins are identifiable and regular, in CMOS hardware.

We agree with the reviewer that the two stochastic bitstreams would be indistinguishable. And we also agree that CMOS systems with a modular developed architecture where physical spins are identifiable do qualify as computing through the natural laws of physics. We have modified the paragraph of the discussion where we address this point in order to highlight that there exist different degrees of abstraction of "spins" and "couplings":

"New learning algorithms grounded in the physics of the hardware are emerging such as Hamiltonian Echo Backprop [72], Coupled Learning [73], Thermodynamics computing [74, 75], Forward-forward-like algorithms [76], Deep reservoir computing [77, 78] and Equilibrium Propagation [42]. Hardware demonstrations of those alternative training algorithms are milestones sought after by the unconventional computing community. By physically implementing the spins and couplings, the hardware, which may utilize a variety of technologies such as CMOS, optics or emerging nanotechnologies [1, 3, 26, 28–41], embodies the algorithm with different degrees of abstraction, instead of relying on highly synthesized and compiled systems."

The authors should make clarifying statements to their paragraph where Wright's paper is cited so that they do not mislead the field into NOT providing comparisons to the types of IMs developed by many groups and companies in the world (Fujitsu, Toshiba, ... etc.).

We have included citations to the Ising machines developed by many groups and companies in the world such as Fujitsu, Toshiba etc in the paragraph where Wright's paper is cited:

"Our work belongs to the growing field of Physical neural networks [26], where the goal is to develop physical systems based on unconventional nanodevices that solve AI tasks through the natural laws of physics [1, 28–30]. We aim to show that a physical system of coupled spins [3, 30–41] can learn to perform supervised AI tasks through an algorithm that harnesses its intrinsic ability to minimize an energy, arising from the natural laws that govern our physical world."

The rest looks good, and after these minor points are taken into account, given the other reviewer suggestions, I am happy to recommend publication.